# Incentivizing Time-Aware Fairness in Data Sharing

**Jiangwei Chen**[1][2]     **Kieu Thao Nguyen Pham**[1]     **Rachael Hwee Ling Sim**[1]
**Arun Verma**[3]     **Zhaoxuan Wu**[3]     **Chuan-Sheng Foo**[2]     **Bryan Kian Hsiang Low**[1]

[1]Department of Computer Science, National University of Singapore, Singapore
[2]Institute for Infocomm Research, Agency for Science, Technology and Research, Singapore
[3]Singapore-MIT Alliance for Research and Technology, Singapore
{chenj,nguyen.pkt,rachael.sim}@u.nus.edu
{arun.verma,zhaoxuan.wu}@smart.mit.edu
foo_chuan_sheng@i2r.a-star.edu.sg
lowkh@comp.nus.edu.sg

## Abstract

In collaborative data sharing and machine learning, multiple parties aggregate their data resources to train a machine learning model with better model performance. However, as the parties incur data collection costs, they are only willing to do so when guaranteed incentives, such as fairness and individual rationality. Existing frameworks assume that all parties join the collaboration simultaneously, which does not hold in many real-world scenarios. Due to the long processing time for data cleaning, difficulty in overcoming legal barriers, or unawareness, the parties may join the collaboration at different times. In this work, we propose the following perspective: As a party who joins earlier incurs higher risk and encourages the contribution from other wait-and-see parties, that party should receive a reward of higher value for sharing data earlier. To this end, we propose a fair and time-aware data sharing framework, including novel time-aware incentives. We develop new methods for deciding reward values to satisfy these incentives. We further illustrate how to generate model rewards that realize the reward values and empirically demonstrate the properties of our methods on synthetic and real-world datasets.

## 1 Introduction

*Collaborative machine learning* (CML) presents an appealing paradigm where multiple parties can aggregate their data to train a *machine learning* (ML) model with better model performance [49, 63]. This paradigm has been widely adopted in various real-world applications, including clinical trials [8, 33, 17], data marketplaces [45, 18], precision agriculture [53, 34]. However, parties, who incur non-trivial data collection and processing costs [22], may be unwilling to collaborate without fair reward. For example, hospitals may be reluctant to share their data with academic institutions due to the substantial investment required to conduct clinical trials and generate medical data [19]. To incentivize such parties to collaborate, existing frameworks [43, 49] have proposed two main steps: In *data valuation*, a party is assigned a value for its shared data. In *reward realization*, a party will receive money, synthetic data, or an ML model as a reward whose value satisfies incentives like *fairness* and *individual rationality* adapted from *cooperative game theory* (CGT). These incentives respectively entail that a party's reward value should be higher than that of others sharing less valuable data [54, 41] and higher than what it can achieve alone.

In this work, we consider how data valuation and reward realization should change if parties join the data sharing process at different times due to the long processing time for data cleaning, delay in overcoming legal barriers, or unawareness of the collaboration opportunities [33] (see App. A for more justification and description of our setting). As a concrete example, hospitals sharing their data

39th Conference on Neural Information Processing Systems (NeurIPS 2025).

from clinical trials often take heterogeneous time to convert these medical data into a standardized format and secure informed consent from their patients [33]. Additionally, a data marketplace mediator, who wishes to encourage participation, may allow parties to freely/asynchronously join the collaboration and continuously update the ML model with new data until a target accuracy is met [1]. Then, the mediator closes the collaboration and rewards the contributing parties.[1] In these examples, **how should a party's reward value change if it joins the collaboration earlier? Should parties contributing data of the same value at different joining times receive equal rewards?**

We propose the following perspective as illustrated by Fig. 1: When parties take different times to join the collaboration, **a party should receive a reward of *higher* value for contributing data earlier to incentivize early contribution.**[2] Firstly, parties who join earlier incur higher risks and hence require rewards of higher value. In the data marketplace example, a party who contributes data early risks receiving late or no reward, unlike another party who contributes just before the target accuracy is met. Next, the early party's contribution may also encourage the contribution from other wait-and-see parties who assess the likelihood of reaching the target accuracy before committing [4]. Our perspective also aligns with the socio-psychological equity theory [16, 47] and the signaling effect [3] observed in economics: Given identical financial contributions, the one contributing earlier is entitled to a reward of higher value.

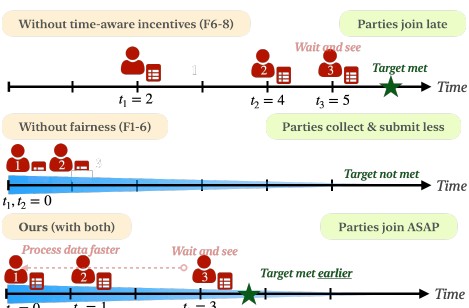

Figure 1: Overview of our data sharing problem setting (App. A) and the impact of fairness and our time-aware incentives (Sec. 4).

Our work addresses two key challenges. The first challenge is to *mathematically formalize time-aware incentives that realize our above perspective*. Naïvely, our perspective would conflict with fairness incentives which dictates that parties contributing data of equal value receive equal rewards *regardless* of their joining times. In Sec. 4, we resolve these conflicts by (i) adding a pre-condition (e.g., equal joining times) such that existing incentives (e.g., fairness) should hold and (ii) only requiring a party's reward value to not decrease (instead of increase) from contributing earlier in some cases. The second challenge is to *decide the reward values that satisfy the existing and new time-aware incentives*. At first glance, one can just divide data value, such as the Shapley value [48, 13], by the joining time. However, this simple solution may overly reduce the reward value of a party who joins late, thus violating *individual rationality*. To tackle this challenge, we introduce two new methods in Sec. 6. Our first **time-aware reward cumulation** method divides the entire duration of collaboration into multiple time intervals and considers each time interval as a separate collaboration among the parties present during that interval. A party's final reward value is a sum of the reward values assigned in each interval, weighted by the time interval importance. Our second method leverages a novel **time-aware data valuation function** and the Shapley value to determine rewards that satisfy all incentives. We then realize the rewards by likelihood tempering [51] and training models only partially on the aggregate data. We empirically validate our proposed time-aware methods in Sec. 7.

## 2 Related Work

**Time Consideration in FL.** *Federated learning* (FL) [38] is a form of collaborative machine learning (and an alternative to data sharing) that involves training on data from multiple parties (clients) without centralizing them. Instead, in each round, collaborating parties share parameters' updates for improving the model (e.g., gradients). The bulk of the FL works [26, 65] has focused on improving the model's utility and operated under the assumption that all parties are fully cooperative and do not require external incentives for contributing their data. While there is some research on providing incentives in FL, such as encouraging collaborative fairness [30, 61] and egalitarian fairness [27, 28], these works do not guarantee that a party's reward (across rounds) improves from contributing data earlier. To provide such guarantees, we consider the data sharing setting as a first step.

---

[1]This time-sensitive setting is realistic as e-commerce marketplaces have used similar techniques (e.g., threshold discounting [36], offering discounts to early customers [42]) to encourage prompt actions [31].

[2]We consider settings where data retain value over time (e.g., medical data), but earlier access enables better utility. We exclude time series data (e.g., stock prices), where recent observations are inherently more predictive.

**Data Valuation.** While existing data valuation methods have employed a range of metrics to value data, such as validation accuracy [21, 13], diversity [64], generalization performance [60], and cost [15], they lack consideration of the parties' joining times (and hence when they contributed their data). [32] can incorporate the parties' joining times based on the assumption that the permutation of parties affects the value of data, but there is no guarantee that a party would receive a reward of higher value for contributing its data earlier. Although data valuation methods in FL can integrate sequential information by considering the weighted average of the parties' contributions in each round (e.g., utilizing uniform weights [57] or decaying weights [52]), they do not explicitly address nor theoretically guarantee individual rationality F2 and time-aware incentives F6 to F8.

**Incentives in CML.** [49, 54, 63] have prescribed how to give out data or model rewards that satisfy incentives (such as fairness) based on the popular cooperative game theory concept, Shapley value [48]. However, these works have also assumed that all parties are present at the start of the collaboration and hence do not realize the perspective that a party should receive a reward of higher value for contributing data earlier. Additionally, while these works suggest how to give out model rewards, our focus is on deciding the reward values for fair and time-aware incentives. [12] is the only other work that studies how to incentivize early arrival in cooperative games. However, our settings, incentives and solutions differ significantly. The key distinctions are discussed at length in App. B.

## 3 Problem Formulation

We consider a problem setting where a trusted mediator (e.g., data-sharing frameworks implemented by government agencies [20]) identifies a common ML task (e.g., disease prediction) of interest to multiple parties (e.g., hospitals). The mediator invites parties to freely/asynchronously join the collaboration by contributing data and continuously updating the ML model with the newly shared data. The mediator closes the collaboration after a target model performance/accuracy is met [1].

Let $n$ denote the number of parties who have joined the collaboration. We denote the set of all parties (i.e., *grand coalition*) as $N \triangleq \{1, \ldots, n\}$ and their respective datasets as $D_1, \ldots, D_n$. Any subset of parties, $C \subseteq N$, is a *coalition* of parties with an aggregated dataset $D_C \triangleq \bigcup_{i \in C} D_i$. In *time-aware* data sharing, we further consider that each party $i$ joins at a different time value $t_i \in \mathbb{Z}_{\geq 0}$ due to differences in processing times, urgency, and risk levels (see App. A for a detailed justification). A larger time value indicates a later joining time, and the party joining earliest is always assigned a time value of $0$.[3] Collectively, the joining time values are denoted by the vector $\boldsymbol{t} \triangleq (t_1, \ldots, t_n)$. After the collaboration closes, the trusted mediator values the shared data, decides the reward value $r_i$, and trains an ML model as a reward (in short, *model reward*) valued at $r_i$ for every party $i \in N$.

A party's data should be valued relative to the data contributed by the other parties [50]. So, to value data, the mediator uses a data valuation function $v$ that maps every coalition $C$ to its value $v_C$. For example, $v_C$ can be the model performance (e.g., validation accuracy) achieved by training on the aggregated dataset $D_C$. We also denote $v_i \triangleq v_{\{i\}}$. As in the works of [49, 30], we do not check the truthfulness of the parties' data and value datasets as-is. Based on the value $v_C$ for each coalition $C \subseteq N$ and the joining time values $\boldsymbol{t}$, the mediator must decide the reward value $r_i$ for every party $i \in N$ that satisfies existing fairness and our time-aware incentives to encourage early contribution. We will outline the incentives in Sec. 4, the necessary characteristics for data valuation function $v$ in Sec. 5, and two new methods to decide reward values $(r_i)_{i \in N}$ that satisfy our incentives in Sec. 6.

## 4 Incentives in Time-Aware Data Sharing

In our time-aware setting, we encourage the parties to share their data as early as possible. To this end, we incorporate existing incentives F1 to F5 (e.g., fairness) from prior works [49] and novelly consider the impact of joining time values when defining time-aware incentives F6 to F8. In this section, we formally define the incentive conditions that should hold for reward values $(r_i)_{i \in N}$ based on the value $v_C$ of aggregated dataset $D_C$ for all coalitions $C \subseteq N$ and the joining time values $\boldsymbol{t}$. We also justify how these incentives encourage parties to join early instead of withholding their data with

---

[3]For clarity, we discretize time into non-negative integers: Given any earliest joining time $t'$ and positive interval length $\ell$, time $t$ maps to $\lfloor (t - t')/\ell \rfloor$. For instance, Windows systems [39] use $t' = 1$ January 1601 00:00 UTC and $\ell = 100 \, \text{ns}$.

a wait-and-see attitude. We use $^*$ to mark incentives where we introduce time-aware considerations and $^\#$ to mark our *new* incentives that specify how $(r_i)_{i \in N}$ should vary for *different* joining times.

F1 **Non-negativity.** The value of reward each party receives should be non-negative: $\forall i \in N \;\; r_i \geq 0$ .

F2 **Individual Rationality.** Each party should receive a reward whose value is at least that of its own data: $\forall i \in N \;\; r_i \geq v_i$ .

F3 **Equal-Time Symmetry$^*$.** If parties $i$ and $j$ enter the collaboration simultaneously, and the inclusion of party $i$'s data results in the same improvement to the model quality as that of party $j$ when trained on the aggregated data of any coalition, then both parties should receive rewards of equal value: $\forall i, j \in N$ s.t. $i \neq j$,

$$(t_i = t_j) \wedge (\forall C \subseteq N \setminus \{i, j\} \;\; v_{C \cup \{i\}} = v_{C \cup \{j\}}) \implies r_i = r_j .$$

F4 **Equal-Time Desirability$^*$.** If parties $i$ and $j$ enter the collaboration simultaneously, and the inclusion of party $i$'s data improves the model quality more than that of party $j$ when trained on the aggregated data of at least one coalition, without the reverse being true, then party $i$ should receive a reward of higher value than party $j$: $\forall i, j \in N$ s.t. $i \neq j$,

$$(t_i = t_j) \wedge (\exists B \subseteq N \setminus \{i, j\} \;\; v_{B \cup \{i\}} > v_{B \cup \{j\}})$$
$$\wedge (\forall C \subseteq N \setminus \{i, j\} \;\; v_{C \cup \{i\}} \geq v_{C \cup \{j\}}) \implies r_i > r_j.$$

F5 **Uselessness$^*$.** If the inclusion of party $i$'s data fails to improve the model quality when trained on the aggregated data of any coalition, then party $i$ is *useless* and should be assigned a reward with no value: $\forall i \in N$,

$$(\forall C \subseteq N \setminus \{i\} \;\; v_{C \cup \{i\}} = v_C) \implies r_i = 0 .$$

In F3 and F4, we add a pre-condition of equal joining time values to accommodate our perspective that a party who joins earlier may receive a higher reward value. However, our adaptation of F5 ignores the joining time values to prevent a useless party from unfairly increasing its reward to $> 0$ by joining earlier, which will disincentivize other parties from sharing data.

F6 **Necessity$^\#$.** If the exclusion of data from either party $i$ or party $j$ renders the model trained on the aggregated data of any remaining coalitions useless, then both parties are *necessary* and should receive rewards of equal value: $\forall i, j \in N$ s.t. $i \neq j$,

$$(\forall C \subseteq N \;\; \{i, j\} \nsubseteq C \implies v_C = 0) \implies r_i = r_j .$$

F6 guarantees that parties essential to the success of the collaboration are treated equally despite having different joining time values and datasets.
*Justification.* A party (e.g., a specialist hospital) may uniquely possess high-quality medical data such that the ML model trained without its data cannot achieve the required accuracy threshold and is untrustworthy and impractical to use [10, 24]. Necessity ensures that a necessary party $j$ is not penalized for joining later. Without necessity, $j$'s reward could decrease to $0$ over time, disincentivizing $j$ from curating high-quality data, joining the collaboration, causing the collaboration to only have value $v_{N \setminus j} = 0$.

F7 **Time-based Monotonicity$^\#$.** When a party $i$ joins the collaboration earlier,[4] *ceteris paribus*, the value of its reward should not decrease. Let $r_i'$ be the new reward received by party $i$ upon the new joining time values $\boldsymbol{t}' \triangleq (t_1', \ldots, t_n')$. Then, $\forall i \in N$,

$$(t_i' < t_i) \wedge (\forall j \in N \setminus \{i\} \;\; t_j' = t_j) \implies r_i' \geq r_i .$$

*Remark* 4.1 (Time-based Equal-Value Desirability$^\#$). A natural consequence of F3 and F7 is time-based desirability. That is, if party $i$ joins the collaboration earlier than party $j$, and party $i$'s data yields the same improvement in model quality as that of party $j$, then the value of reward received by party $i$ should not be less than party $j$: $\forall i, j \in N$ s.t. $i \neq j$,

$$(t_i < t_j) \wedge (\forall C \subseteq N \setminus \{i, j\} \;\; v_{C \cup \{i\}} = v_{C \cup \{j\}}) \implies r_i \geq r_j .$$

To see this, suppose that $t_j' = t_i < t_j$. Then, $r_j' = r_i$ by F3, and $r_i = r_j' \geq r_j$ by F7. This incentive complements and contrasts with the equal-time symmetry F3.

---

[4]The earliest party $j$ is always assigned $t_j = 0$. Thus, its reward value need not improve from joining earlier.

*Remark* 4.2 (Reason for *weak* inequality in F7 and Remark 4.1). F5 requires that any useless party $i$ receive a reward of value 0 despite their joining times, i.e., $r_i' = r_i = 0$. F6 requires that necessary parties $i, j$ receive equal rewards despite $j$ joining earlier. Hence, $r_i' > r_i$ and $r_i > r_j$ would not always hold in F7 and Remark 4.1. Instead, we need an additional pre-condition on party $i$ to define time-based *strict* monotonicity F8:

F8 **Time-based Strict Monotonicity$^{\#}$.** Let $\mathbb{I}_i$ indicate if party $i$'s data yields additional improvement in model quality to that of some coalition comprising only $i$'s predecessors who joined earlier (i.e., $\{j : t_j < t_i\}$). Formally, $\mathbb{I}_i \triangleq (\exists C \subseteq \{j : t_j < t_i\}\ v_{C \cup \{i\}} > v_C + v_i)$. Any party $i$ who joins the collaboration earlier can receive a reward of higher value if $\mathbb{I}_i$ holds: $\forall i \in N$,

$$\mathbb{I}_i \wedge (t_i' < t_i) \wedge (\forall j \in N \setminus \{i\}\ t_j' = t_j) \implies r_i' > r_i .$$

*Remark* 4.3 (Significance of $\mathbb{I}_i$). $\mathbb{I}_i$ guarantees that party $i$ is not useless. When the data valuation function is superadditive (Sec. 5), $\mathbb{I}_i$ guarantees that party $i$'s Shapley value (at an earlier joining time) would be positive, facilitating the guarantee of F8 in Sec. 6.

In this section, we have designed time-aware incentive conditions F6 to F8 while resolving conflicts (e.g., introduce equal time pre-condition in F3 and F4, only require time-based strict monotonicity in certain cases). **A party may get a higher reward from taking time to curate a higher quality dataset instead of sharing a less valuable dataset as early as possible**. This holds in two key scenarios: (i) when a greater emphasis is placed on data quality relative to joining time in the reward schemes; and (ii) when the parties are necessary parties, i.e., their data are essential for achieving non-zero value collaboration. The next step is to design reward schemes (Sec. 6) that satisfy all these incentives. This is non-trivial as existing frameworks fail to satisfy all incentives.

**Failure of the Shapley Value.** The Shapley value [48] is a popular CGT concept that rewards each party $i$ with its average marginal contribution ($v_{C \cup \{i\}} - v_C$) across every possible coalition $C \subseteq N \setminus \{i\}$. Given a utility function $v : 2^N \to \mathbb{R}$ that measures the value of a coalition,

$$\varphi_i(v, N) \triangleq \sum_{C \subseteq N \setminus \{i\}} \frac{|C|!(|N| - |C| - 1)!}{|N|!}(v_{C \cup \{i\}} - v_C) \tag{1}$$

is the Shapley value of $i$ within the grand coalition $N$. When $v$ is time-agnostic (e.g., the accuracy of a model trained on coalition $C$), the Shapley value (and CML frameworks [49, 54] that use it directly) and other weighted sum of marginal contributions [25], fails to satisfy the time-aware incentive F8.

Although dividing the Shapley value by the joining time, i.e., $\varphi_i(v, N)/(t_i + 1)$ satisfies F8, other incentives may be violated. To illustrate, consider a two-party collaboration with $v_1 = v_2 = .2, v_{\{1,2\}} = 1$ and $t_1 = 4$ and $t_2 = 0$. Party 1's time-weighted Shapley value would be $.5/5 = .1 < v_1$, violating F2. As another example, consider $v_1 = v_2 = 0$ instead. The Shapley values are both $.5$ (satisfying F6) but the time-weighted Shapley values are $r_2 = .5 \neq r_1 = .1$, thus violating F6.

## 5 Data Valuation in Time-Aware Data Sharing

To achieve the incentives F1 to F8, it is **sufficient** for the data valuation function $v$ to satisfy:

A1 **Non-negativity.** The value of aggregated data in any coalition is non-negative: $\forall C \subseteq N\ v_C \geq 0$ .

A2 **Monotonicity.** The inclusion of more parties into a coalition does not decrease the value of the aggregated data: $\forall B \subseteq C \subseteq N\ v_C \geq v_B$ .

A3 **Superadditivity.** The value of the aggregated data from two non-overlapping coalitions is no less than the sum of their individual values when the two coalitions are not collaborating: $\forall B, C \subseteq N$ s.t. $B \cap C = \emptyset, v_{B \cup C} \geq v_B + v_C$ .

A1 and A2 align with the norms [50] in CGT and ML, and hold when non-malicious parties contribute valuable data. Crucially, A3 ensures (i) the formation of the grand coalition with *all* parties [5], (ii) the Shapley value satisfies individual rationality (Lemma F.1) and (iii) our methods (Sec. 6) satisfy the time-aware incentives in Sec. 4. A superadditive data valuation function also makes intuitive sense: When parties with less diverse datasets jointly train and deploy a model, the revenue generated by their model with much-improved performance may exceed the total revenue achievable by the parties operating independently. Examples of data valuation function that satisfy A1 to A3 include:

- **Conditional Information Gain (IG).** The conditional IG measures the informativeness of a coalition's data $D_C$, conditioned upon the aggregated data from other parties $D_{-C} \triangleq D_{N \setminus C}$. App. D proves that the conditional IG (Eq. 2) satisfies A1 to A3. The IG [49] (Eq. 3) is the reduction in the uncertainty/entropy of the model parameters $\boldsymbol{\theta}$ after observing $D_C$.

$$v_C \triangleq I(\boldsymbol{\theta}; D_C | D_{-C}) = I(\boldsymbol{\theta}; D_C, D_{-C}) - I(\boldsymbol{\theta}; D_{-C}) \tag{2}$$

$$I(\boldsymbol{\theta}; D_C) \triangleq H(\boldsymbol{\theta}) - H(\boldsymbol{\theta} | D_C) \tag{3}$$

- **Dual of Submodular Valuation Function.** Submodular[5] functions are prevalent in ML [2, 55] and data valuation [50]. For example, IG [49] (3) is monotone submodular. The dual function $v$ of a (submodular) function $v'$ is defined by

$$v(C) \triangleq v'(N) - v'(N \setminus C), \ \forall C \subseteq N , \tag{4}$$

and measures the *importance/contribution of coalition $C$ to the overall collaboration*. In App. E.1, we show that **the dual of any monotone submodular valuation function satisfies A1 to A3**. This connection is significant as the Shapley value for a valuation function and its dual are equal, i.e., $\varphi_i(v', N) = \varphi_i(v, N)$ — our proposed time-aware framework can be used to decide reward values (e.g., monetary payments) and satisfy time-aware incentives for submodular functions.

*Remark* 5.1 (Unlearning Data Valuation). The dual valuation function also admits an interpretation from a machine unlearning perspective. Machine unlearning [62] aims to obtain a model trained on subset of the original training data, excluding the data to be unlearned (e.g., harmful or privacy-sensitive data). The dual data value of $C$ is then the difference in performance between the original model and the model obtained after unlearning $C$. This **performance-drop notion of data value can better capture the indispensability of certain data points**, as these data are often crucial for pushing model performance beyond a high baseline (e.g., from 95% to 98%), whereas most data suffice to reach a decent performance (e.g., from 10% to 70%). Although the dual data value differs from the original data value, as we show in App. E.2, **the Shapley values computed using the dual valuation function and the original valuation function are identical**. This equivalence makes it possible to utilize efficient machine unlearning methods [40] to approximate Shapley-based data values without retraining.

*Remark* 5.2 (Incentives for the Dual). The dual valuation function provides alternative insights into the incentives proposed in Sec. 4. For example, instead of receiving a more valuable reward than a party's original data, individual rationality F2 can be interpreted as a party's reward must be more indispensable (to the grand coalition with value $v'(N)$) than its original data. We defer the relevant proofs and interpretations regarding the dual valuation function to App. E.3.

# 6 Time-Aware Reward Schemes

This section introduces two time-aware methods (Fig. 2) that consider the joining time values $\boldsymbol{t}$. The two methods: *time-aware reward cumulation* (Sec. 6.1) and *time-aware data valuation* (Sec. 6.2) differ on whether the time consideration is introduced after or before using the Shapley value.

## 6.1 Time-Aware Reward Cumulation

To incorporate the time information into the reward distribution stage, we consider each time interval as a separate collaboration among parties present during that interval. Specifically, let $N_\tau \triangleq \{i \in N : t_i \leq \tau\}$ denote the set of parties who join the collaboration before time value $\tau \in \mathbb{Z}_{\geq 0}$. At time value $\tau$, any party $j \notin N_\tau$ is assumed to train a model alone and the collaboration is described by the valuation function restricted to the coalitions in $N_\tau$, that is, $v_{(\cdot)}^{(\tau)} : 2^{N_\tau} \to \mathbb{R}$ such that $v_C^{(\tau)} = v_C, \ \forall C \subseteq N_\tau$.

**Theorem 6.1.** *For each party $i \in N$, its Shapley value at time value $\tau$, $\varphi_i^{(\tau)} \triangleq \varphi_i(v_{(\cdot)}^{(\tau)}, N_\tau)$ if $i \in N_\tau$ and $v_i$ otherwise. Let the weight of time interval $t$ be $w^{(t)} \triangleq \beta^t / \sum_{\tau=0}^T \beta^\tau$ where $\beta > 0$ is a tunable parameter and $T \triangleq \max_{i \in N} t_i$. The reward value $r_i \triangleq \sum_{\tau=0}^T w^{(\tau)} \varphi_i^{(\tau)}$ satisfies F1 to F8.*

The weights are normalized to ensure feasibility (i.e., $r_i \leq v_N \ \forall i \in N$ as the model rewards cannot be better than the model trained on all parties' data). **The mediator can set $\beta$ to control**

---

[5] A valuation function is submodular if $\forall i \in N \ \forall C \subseteq C' \subseteq N \setminus \{i\} \ v_{C' \cup \{i\}} - v_{C'} \leq v_{C \cup \{i\}} - v_C$.

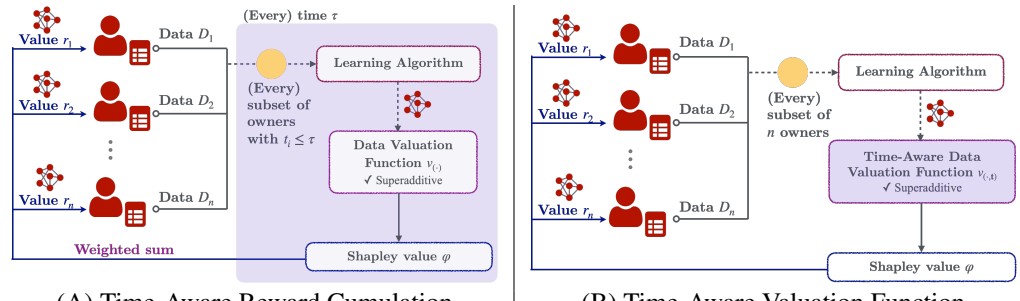

(A) Time-Aware Reward Cumulation   |   (B) Time-Aware Valuation Function

Figure 2: Overview of our proposed methods. (A) We partition the collaboration period into time intervals and consider separate cooperative games for each, rewarding parties via a weighted sum of the corresponding Shapley values (Sec. 6.1). (B) We propose a new time-aware data valuation function and directly use the resulting Shapley values as the reward values (Sec. 6.2).

**the emphasis on joining times and time-aware incentives. As $\beta$ increases, the weight of $\varphi_i^{(\tau)}$ increases for later time intervals; thus, there is less emphasis on the joining times of parties.** Conversely, as $\beta$ decreases, more weight is placed on the earlier time intervals. In fact, as $\beta \to \infty$, the reward value $r_i \to \varphi_i^{(T)}$, which coincides with the Shapley value when all parties have joined, thus no time information is accounted for. The proof of Theorem 6.1 can be found in App. F.1.

*Remark* 6.2 (Efficient Estimation due to Linearity). Instead of computing the Shapley value $T$ times, it is possible to exploit the linearity property of the Shapley value to express $r_i$ as the Shapley value of one valuation function defined on all subsets of $N$: $\nu_C^{(\tau)} \triangleq v_{C \cap N_\tau} + \sum_{j \in C \setminus N_\tau} v_j$, i.e., $r_i \triangleq \varphi_i(\sum_{\tau=0}^{T} w^{(\tau)} \nu_{(\cdot)}^{(\tau)}, N)$. This reduction allows our time-aware reward cumulation method to be combined with other Shapley value approximation methods [46, 21] for efficient estimation. Our time-aware setting only increases the computational complexity by a factor of at most $n$ (the maximum number of unique joining times).

## 6.2 Time-Aware Data Valuation

The next method replaces the data valuation function $v_{(\cdot)}$ with a time-aware data valuation function $v_{(\cdot,\boldsymbol{t})}$ and makes use of propositions from [35] in the CGT literature. [35] has proposed that when $v$ is superadditive and parties have different cooperative levels (denoted by the vector $\boldsymbol{\lambda} \triangleq (\lambda_i)_{i=1}^n \in [0,1]^n$), the Shapley value computed based on

$$v_{C,\boldsymbol{\lambda}} = \sum_{T \subseteq C, |T| \geq 2} d(v,T) \min_{i \in T}\{\lambda_i\} + \sum_{i \in C} d(v, \{i\}) \,,$$

satisfies monotonicity (i.e., as $\lambda_i$ increases, $\varphi_i(v_{(\cdot,\boldsymbol{\lambda})}, N)$ also increases). Here, the Harsanyi dividend [14] $d(v,T) = v_T - \sum_{S \subsetneq T} d(v,S)$ measures the unique contribution of coalition $T$ after removing the contributions of its sub-coalitions [56].[6] We provide a detailed discussion of their results in App. F.2. In our work, we define party $i$'s cooperative level by how early it joins. Formally,

**Theorem 6.3.** *For each party $i \in N$, let party $i$'s cooperative ability $\lambda_i \triangleq e^{-\gamma t_i}$ where $t_i$ is the time value when $i$ joined and $\gamma \in (0,1]$ is a tunable parameter. The time-aware data valuation function is*

$$v_{C,\boldsymbol{t}} \triangleq \sum_{T \subseteq C, |T| \geq 2} d(v,T) \min_{i \in T}\{e^{-\gamma t_i}\} + \sum_{i \in C} d(v, \{i\}) \tag{5}$$

*for all $C \subseteq N$. When $v$ is non-negative and superadditive, $v_{(\cdot,\boldsymbol{t})}$ is also superadditive. Rewarding based on the Shapley value, i.e., $r_i \triangleq \varphi_i(v_{(\cdot,\boldsymbol{t})}, N)$ satisfies F1 to F8.*

**The mediator can set $\gamma$ to control the emphasis on joining times and time-aware incentives. When $\gamma = 0$, the joining time does not matter and every party will have the maximum cooperative level of $1$. As $\gamma$ increases, the joining time has a larger influence on data values.**

---

[6]The dividend is 0 for empty coalitions, i.e., $d(v, \emptyset) = 0$.

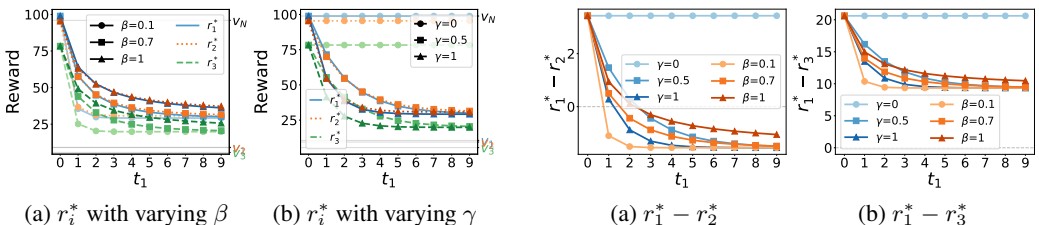

| (a) $r_i^*$ with varying $\beta$ | (b) $r_i^*$ with varying $\gamma$ | (a) $r_1^* - r_2^*$ | (b) $r_1^* - r_3^*$ |

Figure 3: Graphs of $r_i^*$ vs. $t_1$ with the Friedman dataset using methods in (a) Sec. 6.1 (b) Sec. 6.2.

Figure 4: Graphs of differences between reward values with the Friedman dataset.

For $\gamma > 0$, party $i$ only attains the maximum cooperative level of 1 when $t_i = 0$. As $t_i$ increases, party $i$'s cooperative level shrinks towards 0. Thus, the monotonicity property implies that the reward value would decrease. We provide proof of Theorem 6.3 and efficient computation of Eq. (5) in App. F. Similar to the previous approach, this method retains the computational benefits of efficient approximation, and the time-aware setting only increases the complexity by a factor of at most $n$.

### 6.3 Reward Realization

Multiplying the reward values $(r_i)_{i \in N}$ by a positive factor $\rho \geq 1$ still satisfies incentives F1 to F8. This allows us to exploit the freely replicable nature of data by training models with varying qualities to realize values $r_i^* \triangleq \rho r_i$ where $\rho = v_N / \max_{i \in N} \varphi_i(v, N)$. The adjusted values $(r_i^*)_{i \in N}$ ensure that in the time-agnostic case, at least one party is guaranteed to receive a reward equivalent to the best-performing model trained on all aggregated data, i.e., when $t_i = 0 \; \forall i \in N, \exists i \in N \; r_i^* = v_N$ (*weak efficiency* incentive in [49]).

We adopt two approaches to realize the reward values: the *likelihood tempering* method and the *subset selection* method. The former assigns party $i$ a model reward (posterior) $p_i^*$ that is updated using its own likelihood and the tempered likelihood of other parties' data (as in [51]). This method can realize the rewards *exactly* and is suitable when conditional IG is used as the data valuation function. The latter assigns party $i$ a model reward that is only updated on a discrete subset of the aggregated data. Although this latter method can only realize the rewards *approximately*, it is applicable to all data valuation functions. More detailed descriptions of both methods can be found in App. G.

## 7 Experiments and Discussion

This section empirically illustrates the properties of our proposed reward schemes using (a) the synthetic Friedman dataset with 6 input features [11], (b) the *Californian housing* (CaliH) dataset [44] with 8 input features, and (c) the MNIST dataset [7] of handwritten digit images ($28 \times 28$ pixels). We employ the *Gaussian process* (GP) regression [59] model for Friedman and CaliH datasets and *neural network* (NN) for the MNIST dataset. In (a) and (b), each party's data value is measured by conditional IG. In (c), each party's data value is the dual[7] of the validation accuracy, i.e., given the validation accuracy $v'(D)$ of the model trained on $D$, $v(D_i) = v'(D_N) - v'(D_N \setminus D_i)$. Next, each party gets a model reward $p_i^*$ generated by likelihood tempering (a, b) or subset selection (c). We also report the model performance evaluated by *mean negative log probability* (MNLP), defined as $1/|D_{\text{val}}| \cdot \sum_{(\mathbf{x},y) \in D_{\text{val}}} - \log p_i^*(y|\mathbf{x})$ where $D_{\text{val}}$ is the validation set. In (c), MNLP equals to the cross-entropy loss on $D_{\text{val}}$. A *lower* MNLP indicates better model performance. Following [49, 51], we consider $n = 3$ parties in our main paper (and $n = 10$ parties in App. H.6). Party $i$'s reward depends on both its joining time $t_i$ and data value $v_i$. To explore the impact of joining times on rewards, we increase $t_1$ from 0 while keeping $t_2 = t_3 = 0$. For (a) and (b), we indirectly control party $i$'s valuation $v_i$ by varying the number of data points $n_i$. Each party's data $D_i$ is randomly sampled (without replacement) from $D_{\text{train}}$. When all parties draw data uniformly from the same distribution (but do not have sufficient data to achieve the best model performance), the number of data points is positively correlated with the data values, i.e., $n_1 > n_2$ typically leads to $v_1 > v_2$, allowing us to demonstrate desirability (F4). For (c), we vary each party's data value by restricting the labels of their data. **For baselines, we compare against the Shapley value, the standard approach in**

---

[7]As the accuracy is approximately submodular, its dual satisfies the properties outlined in Sec. 5.

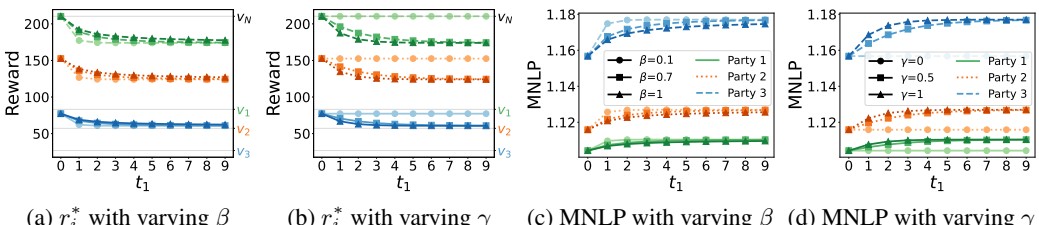

(a) $r_i^*$ with varying $\beta$    (b) $r_i^*$ with varying $\gamma$    (c) MNLP with varying $\beta$   (d) MNLP with varying $\gamma$

Figure 5: Graphs of (a, b) reward values and (c, d) MNLP vs. joining time $t_1$ on the CaliH dataset.

**incentive-aware CML without incorporating joining time information** [49, 43]. It is a special case of our methods when $\gamma = 0$ for time-aware data valuation and $\beta \to \infty$ for time-aware reward accumulation and correspond to horizontal lines in our experimental results (e.g., Fig. 3b)—Shapley value-based rewards remain constant regardless of joining time and do not satisfy incentive F8.

**Overview of Observations.** All experiments using both reward schemes show the satisfaction of incentives: (i) All parties benefit from collaboration and receive rewards more valuable than their own data (F2). (ii) When a party delays its participation, *ceteris paribus*, it receives a model with worse performance (F8). In the time-agnostic case, (iii) parties with higher Shapley values $\varphi_i$ (usually reflected as higher data values $v_i$) receive models with better performance (F4) and (iv) there is always one party who receives the best model with value $v_N$ (weak efficiency).

**Friedman Dataset ($n_1 = n_2 > n_3$).** In Figs. 3-6, we sample parties' data from the Friedman dataset, creating a scenario where $v_1 = 10.58$ is close to $v_2 = 9.43$, while $v_3 = 5.48$ is significantly smaller. When $t_1 = t_2 = t_3 = 0$, the Shapley values are $\varphi_1 = 35.88, \varphi_2 = 34.64, \varphi_3 = 28.40$, with party 1 receiving model with the highest possible value $v_N$, as seen in Fig. 3. As party 1 joins later ($t_1$ increases), Fig. 3 shows that its reward $r_1^*$ decreases as a disincentive for joining late. While F8 only stipulates a decrease in $r_1^*$, we also see a drop in $r_2^*$ and $r_3^*$. This is because other parties receive benefits from party 1's collaboration for a shorter duration, resulting in lower total rewards. However, each party is still guaranteed individual rationality (F2), i.e., all parties receive rewards at least as valuable as their own data (plotted as grey horizontal lines in Fig. 3).

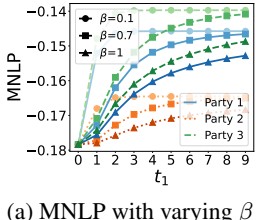

(a) MNLP with varying $\beta$

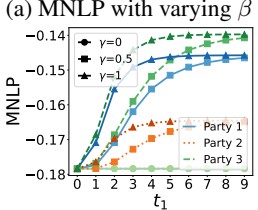

(b) MNLP with varying $\gamma$

Figure 6: Graphs of MNLP vs. $t_1$ on Friedman dataset using (a) time-aware reward cumulation and (b) time-aware data valuation.

Next, we investigate the impact of a party's joining time value (e.g., $t_1$) on the difference of its reward value with others (e.g., $r_1^* - r_2^*$ and $r_1^* - r_3^*$). When the gap between parties' data values are small and $\gamma > 0$[8], the effect of the joining time values is dominant. Although party 1 possesses more valuable data ($v_1 > v_2$), in Fig. 4a, we observe that party 1 receives a lower reward than party 2 ($r_1^* - r_2^* < 0$) when it joins too late. When $\beta = 1$ and $\gamma = 0.5$, party 1 receives lower reward than party 2 if it joins at $t_1 \geq 3$. When $\beta = 0.7$ and $\gamma = 1$, party 1 receives lower reward if it joins at $t_1 \geq 2$ and $r_1^* - r_2^*$ decreases faster. Thus, we observe that using a smaller $\beta$ or a larger $\gamma$ will increase the emphasis on earlier participation and time-based desirability.

However, when there is a significant gap between the data values of two parties (e.g., $v_3 \ll v_1$), Fig. 4b shows that party 1 would always receive a higher reward than party 3 despite joining later (i.e., $r_1^* - r_3^* > 0$). Thus, our time-aware framework balances the consideration of data values and time values and encourages all parties to both curate high-quality data and join earlier to receive higher rewards. We further verify in Fig. 6 that models with higher reward values $r_i$ also have better predictive performance.

**CaliH Dataset ($n_1 > n_2 > n_3$).** We construct a different scenario with CaliH dataset with significant gaps between the data values ($v_1 = 83.15, v_2 = 57.17, v_3 = 26.86$). We report conditional IG and MNLP of the received rewards using both reward schemes in Fig. 5. We observe that the value of

---

[8]When $\gamma = 0$, $v_{C,t} = v_C$. The reward values of all parties are time-agnostic and constant across the time values, resulting in the horizontal $\gamma = 0$ lines in Figs. 3-4.

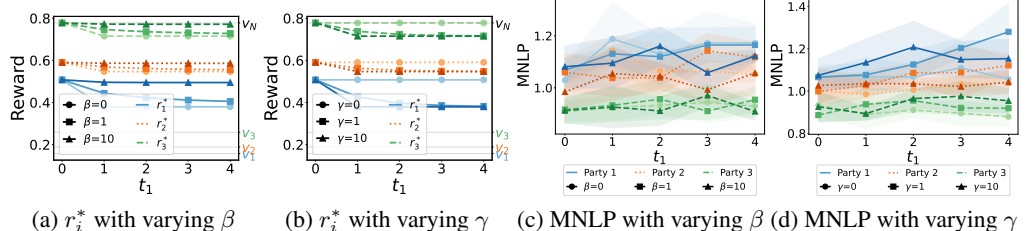

(a) $r_i^*$ with varying $\beta$    (b) $r_i^*$ with varying $\gamma$    (c) MNLP with varying $\beta$ (d) MNLP with varying $\gamma$

Figure 7: Graphs of (a, b) reward values $r_i^*$ and (c, d) MNLP (averaged over 6 independent realization with standard deviation shaded) vs. different joining time value $t_1$ on the MNIST dataset.

rewards always exceeds the value of each party's own data. In addition, the model performance of party 1 decreases as it joins later. These observations are in line with incentives F2 and F8. Lastly, there is no change in the ranking of each party's rewards across all joining times, as the large gaps between the data values outweigh the benefits of earlier participation.

**MNIST Dataset.** Each party has access only to data with a limited subset of labels (see App. H) and has data values $v_1{=}0.16, v_2{=}0.19, v_3{=}0.26$. Fig. 7 reports assigned rewards and reward model performance under both reward schemes. Our reward values adhere to the desiderata in Sec. 4: all assigned rewards outperform individual values (F2), and rewards decrease when joining later (F8). Reward model performance generally declines with later joining times, and party 3, receiving the highest reward, achieves the lowest MNLP, thereby penalizing late joiners and encouraging high-quality data curation. However, the trend is not strictly monotonic, likely due to approximation errors in subset selection and randomness in NN training. Nevertheless, this does not contradict our theoretical results. Our theoretical results are only for the reward values (Fig. 7a-b) and in Fig. 7c-d, we are optionally examining the impact on MNLP (another measure of model performance) when using the subset selection reward realization mechanism.

## 8   Conclusion

This paper seeks to encourage parties to join data sharing collaboration early and curate high-quality data. To this end, we define time-aware incentives that complement existing fairness incentives and propose two time-aware reward schemes that satisfy all incentives, and have parameters to control the emphasis on joining times and earlier participation. Our empirical evaluations show that the incentives hold with respect to the data valuation function and the model predictive performance. We discuss the limitations of our work in App. C, focusing on computational efficiency and privacy.

## Acknowledgments and Disclosure of Funding

This research/project is supported by the National Research Foundation, Singapore under its AI Singapore Programme (AISG Award No: AISG3-RP-2022-029). This research is supported by the National Research Foundation (NRF), Prime Minister's Office, Singapore under its Campus for Research Excellence and Technological Enterprise (CREATE) programme. The Mens, Manus, and Machina (M3S) is an interdisciplinary research group (IRG) of the Singapore MIT Alliance for Research and Technology (SMART) centre. Jiangwei Chen is supported by the Institute for Infocomm Research (I²R), Agency for Science, Technology and Research (A*STAR). We would like to thank the anonymous reviewers and the AC for their helpful and constructive feedback.

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

# A    Further Justification on the Time-Aware Data Sharing Setting

A party may not want or be able to share their data early because:

- *They may be waiting for confirmation that the collaboration benefits them.*
  In footnote 1, we describe how e-commerce marketplaces restrict opportunities to customers to create urgency and encourage prompt actions. Similarly, a data sharing mediator may announce that it will try to collect data but will only give out model rewards if the model has good enough performance (e.g., 90% accuracy) such that it is worth the effort of all parties and extra costs (e.g., legal fees). The mediator may regularly update parties about the current performance and a wait-and-see party might only start contributing when the accuracy is near 80%.

- *They may take time to process the data and the time can be sped up.*
  For example, an incentivized healthcare firm may be proactive in seeking consent from their patients, legal consent, and anonymizing it.

- *They may take more time to collect more data.*
  In situations where longer time means more data collection, the healthcare firm can submit multiple datasets and submit each dataset as soon as possible to maximize its reward, i.e., it becomes multiple parties. Our framework can also be modified to support the case where it is just one party. We are not incentivizing smaller datasets over larger ones. Instead, our goal is to incentivize submission as soon as available.

Thus, we design incentives to **incentivize each party to share their data as early as possible**. Note that while we incentivize parties to share the same dataset as early as possible, our consideration of *fairness, F6 and data valuation* ensures that we do **not** incentivize parties to submit smaller, lower quality, less diverse datasets to join earlier in the collaboration. Our experiments (e.g., Fig. 4b) also empirically demonstrate that parties with significantly more valuable data will still receive better rewards despite joining late.

While it may seem more practical to consider the online **federated learning** setting with repeated data sharing, data valuation, and reward allocation, it may not be possible to ensure our strong theoretical properties on the reward values. Thus, we consider **one-time data sharing** as a first step and leave federated learning to future work.

[49] described some realistic use cases of data sharing. In precision agriculture, a farmer with limited land area and sensors can combine his collected data with the other farmers to improve the modeling of the effect of various influences (e.g., weather, pest) on his crop yield. In real estate, a property agency can pool together its limited transactional data with that of the other agencies to improve the prediction of property prices. Additionally, we note that firms and other parties have and will be willing to share their data with one another using secure sharing platforms (such as X-road [https://x-road.global] which is currently implemented in over 20 countries) which prevent data from being accessed by unauthorized individuals.

It is also worth noting that the non-centralized training (e.g., federated learning), while appealing, does not strictly guarantee privacy either. For example, [58] shows that "private information can still be divulged by analyzing uploaded parameters from clients". Hence, the challenge of preserving privacy is non-trivial in both data-sharing collaboration (CML) and non-data-sharing collaboration (FL), and lies outside the scope of our contributions.

# B    Key Differences with [12]

**Settings.** [12] considers the *online* setting (i.e., distributing rewards every time a party joins), whereas we consider the *offline* setting (i.e., distributing rewards after all parties join). The online setting poses challenges in adjusting rewards to ensure overall fairness if one party's data value changes when additional parties join, as it is difficult to claw back rewards that have already been distributed. In contrast, we focus on the offline setting, where reward distribution occurs only after the collaboration terminates under a predetermined condition (Sec. 3), ensuring fair evaluation and compensation.

Additionally, we consider the *differences in joining times* while [12] considers only the *order or permutation of arrivals*. Our setting is more realistic since in CML, a day's and a month's wait have

different impacts. We introduce this notion by discretizing joining time values (Sec. 3) and propose equal-time incentives F3, F4 and strict time monotonicity F8 (Sec. 4).

**Incentives.** [12] enforces *online individual rationality* (OIR) which ensures that parties' rewards will not decrease when new parties join. However, this could be unfair in CML and other data valuation applications. For instance, if the validation accuracy is used as the valuation function, the first contributor could receive credit for the majority of the accuracy improvement, say, from 0 to 0.8, and this credit would not decrease under OIR, even if a later contributor with more valuable data increases the accuracy from 0.8 to 0.99. In this scenario, a fairer evaluation would reduce the first contributor's attributed contribution. However, in the online setting, as previously discussed, there is no mechanism to retrieve rewards that have already been distributed.

Instead, we build upon fairness incentives from existing CML literature to incorporate joining time values when defining *time-aware incentives* (Sec. 4), aiming to encourage both early participation and the curation of high-quality data. Specifically, the method proposed in [12] fails to satisfy our F3, F4, F6 and F8 incentives.

**Solutions.** Our solutions incentive early participation across *a broader range of valuation functions*. We outline the necessary properties of the valuation function to satisfy all of our proposed incentives and provide a sufficient example (conditional IG) in Sec. 5. In contrast, Corollary 5.7 in [12] indicates that their solution is restricted to *symmetric monotone valuation functions* (i.e., $v(S) = v(T)$ if $|S| = |T|$, and $v(S) \leq v(T)$ if $S \subseteq T$), which significantly limits its practical applicability.

# C   Limitations

**Computational Efficiency.** One limitation of our work is that the exact computation of the proposed methods requires enumerating all possible subsets of the grand coalition, resulting in exponential computational complexity. However, the scalability challenge is **not unique to this work**; It applies to any approach [54, 32, 13, 50] that computes data values using the Shapley value. The only difference is that, to incorporate the additional joining time information, our proposed methods increase the computational complexity **at most linearly**, i.e., by the maximum number of unique joining times, $n$.

This challenge **can be avoided or mitigated** for two reasons. Firstly, in our collaborative data sharing and machine learning setting [49], there are are fewer participating parties, each contributing higher-quality data sources (e.g., hospitals holding larger datasets). Secondly, as discussed in Remark 6.2 and App. F.4, our theoretical results would also hold for unbiased Shapley value estimators, allowing the use of any Shapley value approximation method [21]. On average, the approximate Shapley values still incentivize parties to curate high-quality data and join the collaboration early. Using approximate Shapley values would significantly reduce the number of coalitions to evaluate to $O(n \log n)$ and scalability would also improve with the discovery of more efficient approximation techniques [29]. We evaluate on 10 parties in App. H.6 using Monte Carlo approximation methods and demonstrate that our results remain consistent with an increased number of parties and Shapley value estimation.

**Privacy Issues of Non-FL Setting.** Another limitation of our work is that data sharing poses privacy risks due to the sensitive nature of the data. While it may seem more practical to consider the online *federated learning* (FL) setting with repeated data sharing, data valuation, and reward allocation, it may not be possible to ensure our strong theoretical properties on the reward values. Therefore, we consider **one-time data sharing** as a first step and leave the extension to FL for future work.

It is also important to note that the non-centralized training (e.g., FL), while appealing, does not strictly guarantee privacy either. For example, [58] shows that "private information can still be divulged by analyzing uploaded parameters from clients". Hence, the challenge of preserving privacy is non-trivial in both data-sharing collaboration (CML) and non-data-sharing collaboration (FL), and lies outside the scope of our contributions.

Additionally, we note that firms and other parties have and will be willing to share their data with one another using secure sharing platforms such as X-road (`https://x-road.global`), which is currently deployed in over 20 countries and prevents unauthorized access to shared data.

# D    Proof of Characteristics of Conditional IG

To ease notations, we define $v(\cdot) : 2^N \to \mathbb{R}$ by $v(A) = v_A \ \forall A \subseteq N$ where $v_A = I(\boldsymbol{\theta}; D_A | D_{-A})$ is the conditional IG (2). We need to show that $v$ is non-negative, superadditive, and monotonic.

- **A1 Non-negativity.** The non-negativity of $v$ is apparent from the fact that conditional mutual information is never negative [6].

- **A3 Superadditivity.** We need to show $\forall A, B \subseteq N$ such that $A \cap B = \emptyset$,

$$
\begin{aligned}
v(A \cup B) &= I(\boldsymbol{\theta}; D_{A \cup B} | D_{-(A \cup B)}) \\
&\geq I(\boldsymbol{\theta}; D_A | D_{-A}) + I(\boldsymbol{\theta}; D_B | D_{-B}) \\
&= v(A) + v(B) .
\end{aligned}
$$

By chain rule of mutual information, $I(\boldsymbol{\theta}; D_{A \cup B} | D_{-(A \cup B)}) = I(\boldsymbol{\theta}; D_A | D_{-(A \cup B)}) + I(\boldsymbol{\theta}; D_B | D_{-B})$. It is then sufficient to show

$$
I(\boldsymbol{\theta}; D_A | D_{-(A \cup B)}) \geq I(\boldsymbol{\theta}; D_A | D_{-A}) , \tag{6}
$$

which is equivalent to

$$
H(D_A | D_{-(A \cup B)}) - H(D_A | \boldsymbol{\theta}, D_{-(A \cup B)}) \geq H(D_A | D_{-A}) - H(D_A | \boldsymbol{\theta}, D_{-A})
$$

by the definition of conditional mutual information. If we assume[9] that $D_A$ is conditionally independent of $D_{-A}$ given $\boldsymbol{\theta}$, then

$$
H(D_A | \boldsymbol{\theta}, D_{-(A \cup B)}) = H(D_A | \boldsymbol{\theta}) = H(D_A | \boldsymbol{\theta}, D_{-A}) .
$$

Finally, $H(D_A | D_{-(A \cup B)}) \geq H(D_A | D_{-A})$ since additional information cannot increase uncertainty (entropy). Thus, $I(\boldsymbol{\theta}; D_A | D_{-(A \cup B)}) \geq I(\boldsymbol{\theta}; D_A | D_{-A})$. We have successfully shown (6), which concludes the proof.

- **A2 Monotonicity.** We need to show $\forall B \subseteq C \subseteq N \ \ v(B) \leq v(C)$. Let $A = C \backslash B$, then $A \cap B = \emptyset$ and $A \cup B = C$. By the superadditivity of $v$, $v(C) = v(B \cup A) \geq v(B) + v(A) \geq v(B)$, where the last inequality is because $v(A) \geq 0$.

# E    Properties of Dual Valuation Function

## E.1    Dual of a Monotonic and Submodular Function

We show that the dual of the monotonic, submodular function $v'$ is non-negative, monotonic, and superadditive.

- **A1 Non-negativity.** $\forall C \subseteq N \ \ v(C) = v'(N) - v'(N \setminus C) \geq 0$ where the inequality is due to the monotonicity of $v'$.

- **A2 Monotonicity.** $\forall B \subseteq C \subseteq N \ \ v(C) = v'(N) - v'(N \setminus C) \geq v'(N) - v'(N \setminus B) = v(B)$ since $v'(N \setminus C) \leq v'(N \setminus B)$ by the monotonicity of $v'$.

- **A3 Superadditivity.** By the definition of submodularity, for any $C, D \subseteq N$, $v'(C \cup D) + v'(C \cap D) \leq v'(C) + v'(D)$. For any $A, B \subseteq N$ such that $A \cap B = \emptyset$, we can choose $C = N \setminus A$, $D = N \setminus B$, so $C \cup D = N$ since $A \cap B = \emptyset$ and $C \cap D = N \setminus (A \cup B)$. Therefore,

$$
\begin{aligned}
v'(C \cup D) + v'(C \cap D) &\leq v'(C) + v'(D) \\
v'(N) + v'(N \setminus (A \cup B)) &\leq v'(N \setminus A) + v'(N \setminus B) \\
v'(N) - v'(N \setminus A) - v'(N \setminus B) &\leq -v'(N \setminus (A \cup B)) \\
v'(N) - v'(N \setminus A) + v'(N) - v'(N \setminus B) &\leq v'(N) - v'(N \setminus (A \cup B)) \\
v(A) + v(B) &\leq v(A \cup B) ,
\end{aligned}
$$

proving the superadditivity of $v$.

---

[9] This assumption is also made in [49].

## E.2 Equivalence of Dual when Calculating the Shapley Value

For a dual valuation function $v$, its Shapley value is calculated by

$$\varphi_i(v, N) = \sum_{C \subseteq N \setminus \{i\}} \frac{|C|!(|N| - |C| - 1)!}{|N|!} (v(C \cup \{i\}) - v(C))$$

$$= \sum_{C \subseteq N \setminus \{i\}} \frac{|C|!(|N| - |C| - 1)!}{|N|!} (v'(N) - v'(N \setminus (C \cup \{i\})) - v'(N) + v'(N \setminus C))$$

$$= \sum_{C \subseteq N \setminus \{i\}} \frac{|C|!(|N| - |C| - 1)!}{|N|!} (v'(N \setminus C) - v'(N \setminus (C \cup \{i\})))$$

$$= \varphi_i(v', N)$$

where the last equality is because of the bijection between the sets $\{C : C \subseteq N \setminus \{i\}\}$ and $\{N \setminus (C \cup \{i\}) : C \subseteq N \setminus \{i\}\}$.

## E.3 Dual Interpretations of Incentives

As we define the incentives in Sec. 4 with a superadditive valuation function $v$, which can be the dual of a submodular valuation function $v'$, we investigate the interpretations of these incentives by expressing $v$ in terms of $v'$ in the definitions. We illustrate this using the *subset selection* reward realization method as it is applicable to dual of any submodular functions. Generally, the subset selection method assigns a model trained on $D_i \subseteq D_i^r \subseteq D_N$ as reward to party $i$ such that $v(D_i^r) = r_i^*$ (see App. G for more details). Remarkably, we find that F3 to F5 are equivalent to the corresponding incentives defined in terms of $v'$ while alternative insights can be drawn from interpreting F2 and F6 using the original submodular valuation function $v'$.

- F2 **Individual Rationality.** By the definition of the dual valuation function (4), the condition of F2 translates to

$$v(D_i^r) \geq v(D_i)$$
$$v'(N) - v'(N \setminus D_i^r) \geq v'(N) - v'(N \setminus D_i)$$
$$v'(N \setminus D_i) \geq v'(N \setminus D_i^r) .$$

  This translation suggests that the value (using the original submodular valuation function) of the grand coalition without party $i$'s reward is less than the value of the grand coalition without party $i$'s original data. Therefore, the dual interpretation of F2 is that with the assigned reward, party $i$'s importance/indispensability to the overall collaboration is increased as compared to its importance without the reward.

- F3 **Equal-Time Symmetry.** The condition in F3 can be translated to $\forall C \subseteq N \setminus \{i, j\}$,

$$v(C \cup \{i\}) = v(C \cup \{j\})$$
$$v'(N) - v'(N \setminus (C \cup \{i\})) = v'(N) - v'(N \setminus (C \cup \{j\}))$$
$$v'(N \setminus (C \cup \{j\})) = v'(N \setminus (C \cup \{i\})) .$$

  For any $C \subseteq N \setminus \{i, j\}$, we can take $C' = N \setminus C \setminus \{i, j\} \subseteq N \setminus \{i, j\}$ so that $N \setminus (C' \cup \{j\}) = C \cup \{i\}$ and $N \setminus (C' \cup \{i\}) = C \cup \{j\}$. Since we are considering all $C \subseteq N \setminus \{i, j\}$, the dual interpretation of F3 is equivalent to the original submodular case.

- F4 **Equal-Time Desirability.** The condition of F4 holds when $\exists B \subseteq N \setminus \{i, j\} \; v(B \cup \{i\}) > v(B \cup \{j\})$ and $\forall C \subseteq N \setminus \{i, j\} \; v(C \cup \{i\}) \geq v(C \cup \{j\})$. Translating the inequality after the first quantifier:

$$v(B \cup \{i\}) > v(B \cup \{j\})$$
$$v'(N) - v'(N \setminus (B \cup \{i\})) > v'(N) - v'(N \setminus (B \cup \{j\}))$$
$$v'(N \setminus (B \cup \{j\})) > v'(N \setminus (B \cup \{i\})) .$$

  If we can find $B \subseteq N \setminus \{i, j\}$ such that $v(B \cup \{i\}) > v(B \cup \{j\})$, we can also find $B' \subseteq N \setminus \{i, j\}$ such that $v'(B' \cup \{i\}) > v'(B' \cup \{j\})$ by taking $B' = N \setminus B \setminus \{i, j\}$. Similarly, we can show equivalence between dual and original valuation functions for the condition after the second quantifier. Thus, the dual interpretation of F4 is equivalent to the original submodular case.

- **F5 Uselessness.** A party is useless if $\forall C \subseteq N \setminus \{i\}\ \ v(C \cup \{i\}) = v(C)$. By the definition of dual, $v(C \cup \{i\}) = v(C) \implies v'(N \setminus C) = v'(N \setminus (C \cup \{i\}))$. Noting the bijection between the sets $\{C : C \subseteq N \setminus \{i\}\}$ and $\{N \setminus (C \cup \{i\}) : C \subseteq N \setminus \{i\}\}$, so a useless party also satisfies $\forall C \subseteq N \setminus \{i\}\ \ v'(C \cup \{i\}) = v'(C)$.

- **F6 Necessity.** A party is necessary if $\forall C \subseteq N\ \ \{i,j\} \not\subseteq C \implies v(C) = 0$, i.e., $v'(N) = v'(N \setminus C)$. It suggests that if either $i$ or $j$ is absent from a coalition $C$, then $C$ is of no importance to the overall collaboration as it makes no difference when excluding $C$ from the collaboration. Hence, both $i$ and $j$ are vital for the importance of a coalition and should be assigned the same reward.

# F  Proofs of Theorems

We first prove some useful lemmas regarding the properties of the Shapley value (1).

**Lemma F.1.** *In a grand coalition $N$, if $v$ is superadditive, then $\varphi_i(v, N) \geq v_i$, for all $i \in N$.*

*Proof.* The proof follows directly from the definition of the Shapley value (1) and the superadditivity of $v$:

$$\varphi_i(v, N) = \sum_{C \subseteq N \setminus \{i\}} \frac{|C|!(|N| - |C| - 1)!}{|N|!}(v_{C \cup \{i\}} - v_C)$$

$$\geq \sum_{C \subseteq N \setminus \{i\}} \frac{|C|!(|N| - |C| - 1)!}{|N|!} \cdot v_i$$

$$\geq v_i \, . \qquad \square$$

**Corollary F.2.** *If $v$ is superadditive and non-negative, then $\forall i \in N\ \ \varphi_i(v, N) \geq 0$.*

*Proof.* Following Lemma F.1: $\varphi_i(v, N) \geq v_i \geq 0$. $\qquad \square$

**Lemma F.3** (Necessity). *The Shapley value satisfies the necessity property, i.e., for all $i, j \in N$ such that $i \neq j$, if $\forall C \subseteq N\ \ \{i,j\} \not\subseteq C \implies v_C = 0$, then $\varphi_i(v, N) = \varphi_j(v, N)$.*

*Proof.* Suppose $i$ and $j$ are both necessary parties. We split the calculation of $\varphi_i(v, N)$ into coalitions that do not contain $j$ and coalitions that contain $j$:

$$\varphi_i(v, N)$$

$$= \sum_{C \subseteq N \setminus \{i\}} \frac{|C|!(|N| - |C| - 1)!}{|N|!}(v_{C \cup \{i\}} - v_C)$$

$$= \sum_{C \subseteq N \setminus \{i,j\}} \frac{|C|!(|N| - |C| - 1)!}{|N|!} \underbrace{(v_{C \cup \{i\}} - v_C)}_{A} + \sum_{C \subseteq N \setminus \{i,j\}} \frac{(|C| + 1)!(|N| - |C| - 2)!}{|N|!} \underbrace{(v_{C \cup \{i,j\}} - v_{C \cup \{j\}})}_{B}$$

$$= \sum_{C \subseteq N \setminus \{i,j\}} \frac{(|C| + 1)!(|N| - |C| - 2)!}{|N|!} \cdot v_{C \cup \{i,j\}}$$

where the last equality is because $A = 0$ and $B = v_{C \cup \{i,j\}}$. As $i$ and $j$ are necessary parties, for any coalition $C \subseteq N \setminus \{i,j\}$, $v_C = v_{C \cup \{i\}} = v_{C \cup \{j\}} = 0$. Similarly,

$$\varphi_j(v, N)$$

$$= \sum_{C \subseteq N \setminus \{j\}} \frac{|C|!(|N| - |C| - 1)!}{|N|!}(v_{C \cup \{j\}} - v_C)$$

$$= \sum_{C \subseteq N \setminus \{i,j\}} \frac{|C|!(|N| - |C| - 1)!}{|N|!}(v_{C \cup \{j\}} - v_C) + \sum_{C \subseteq N \setminus \{i,j\}} \frac{(|C| + 1)!(|N| - |C| - 2)!}{|N|!}(v_{C \cup \{i,j\}} - v_{C \cup \{i\}})$$

$$= \sum_{C \subseteq N \setminus \{i,j\}} \frac{(|C| + 1)!(|N| - |C| - 2)!}{|N|!} \cdot v_{C \cup \{i,j\}} \, ,$$

which is exactly the same as $\varphi_i(v, N)$. $\qquad \square$

It is also well known that the Shapley value satisfies the symmetry, uselessness [5] and desirability [37] properties. For ease of reading and exposition, we list these results as lemmas below:

**Lemma F.4** (Symmetry). *The Shapley value satisfies the symmetry property, i.e., for all $i, j \in N$ such that $i \neq j$, if $\forall C \subseteq N \setminus \{i,j\}$ $v_{C \cup \{i\}} = v_{C \cup \{j\}}$, then $\varphi_i(v, N) = \varphi_j(v, N)$.*

**Lemma F.5** (Uselessness). *The Shapley value satisfies the uselessness property, i.e., for all $i \in N$, if $\forall C \subseteq N \setminus \{i\}$ $v_{C \cup \{i\}} = v_C$, then $\varphi_i(v, N) = 0$.*

**Lemma F.6** (Desirability). *The Shapley value satisfies the desirability property, i.e., for all $i, j \in N$ such that $i \neq j$, if $\forall C \subseteq N \setminus \{i,j\}$ $v_{C \cup \{i\}} \geq v_{C \cup \{j\}}$, then $\varphi_i(v, N) \geq \varphi_j(v, N)$. Further, if $\forall C \subseteq N \setminus \{i,j\}$ $v_{C \cup \{i\}} \geq v_{C \cup \{j\}}$ and $\exists B \subseteq N \setminus \{i,j\}$ $v_{B \cup \{i\}} > v_{B \cup \{j\}}$, then $\varphi_i(v, N) > \varphi_j(v, N)$.*

## F.1 Proof of Theorem 6.1

Since we are considering each time interval $\tau$ as a separate collaboration in which a new valuation function $v_{(\cdot)}^{(\tau)}$ is defined, we first note a crucial observation that $v_{(\cdot)}^{(\tau)}$ preserves the non-negativity and superadditivity of $v$:

**Lemma F.7.** *If $v : 2^N \to \mathbb{R}$ is non-negative (superadditive), then $v_{(\cdot)}^{(\tau)} : 2^{N_\tau} \to \mathbb{R}$ as defined in Sec. 6.1 is non-negative (superadditive) for all $0 \leq \tau \leq T$ where $T = \max_{i \in N} t_i$.*

*Proof.* This is clear as we define $v_{(\cdot)}^{(\tau)}$ to be $v$ restricted to the coalitions in $N_\tau$. For any $0 \leq \tau \leq T$, we have

$$v_C^{(\tau)} = v_C \geq 0 \quad \forall C \subseteq N_\tau \subseteq N \,,$$

and

$$v_{B \cup C}^{(\tau)} = v_{B \cup C} \geq v_B + v_C = v_B^{(\tau)} + v_C^{(\tau)} \,,$$

for all $B, C \subseteq N_\tau \subseteq N$ s.t. $B \cap C = \emptyset$. $\qquad\square$

- F1 **Non-negativity.** This follows from F2 and non-negativity of $v$.
- F2 **Individual Rationality.** For all $i \in N$,

$$
\begin{aligned}
r_i &= \sum_{\tau=0}^{T} w^{(\tau)} \varphi_i^{(\tau)} \\
&= \sum_{\tau=0}^{t_i - 1} w^{(\tau)} \varphi_i^{(\tau)} + \sum_{\tau=t_i}^{T} w^{(\tau)} \varphi_i^{(\tau)} \\
&= \sum_{\tau=0}^{t_i - 1} w^{(\tau)} v_i + \sum_{\tau=t_i}^{T} w^{(\tau)} \varphi_i \left( v_{(\cdot)}^{(\tau)}, N_\tau \right) \\
&\geq \sum_{\tau=0}^{t_i - 1} w^{(\tau)} v_i + \sum_{\tau=t_i}^{T} w^{(\tau)} v_i \\
&= \sum_{\tau=0}^{T} w^{(\tau)} v_i \\
&= v_i
\end{aligned}
$$

where the inequality is due to Lemma F.1 and Lemma F.7.

- **F3 Equal-Time Symmetry.** For all $i, j \in N$ s.t. $i \neq j$, if $t_i = t_j$ and $\forall C \subseteq N \setminus \{i, j\}$ $v_{C \cup \{i\}} = v_{C \cup \{j\}}$, then

$$
\begin{aligned}
r_i &= \sum_{\tau=0}^{t_i - 1} w^{(\tau)} \varphi_i^{(\tau)} + \sum_{\tau=t_i}^{T} w^{(\tau)} \varphi_i^{(\tau)} \\
&= \sum_{\tau=0}^{t_i - 1} w^{(\tau)} v_i + \sum_{\tau=t_i}^{T} w^{(\tau)} \varphi_i \left( v_{(\cdot)}^{(\tau)}, N_\tau \right) \\
&= \sum_{\tau=0}^{t_j - 1} w^{(\tau)} v_j + \sum_{\tau=t_j}^{T} w^{(\tau)} \varphi_j \left( v_{(\cdot)}^{(\tau)}, N_\tau \right) \\
&= \sum_{\tau=0}^{t_j - 1} w^{(\tau)} \varphi_j^{(\tau)} + \sum_{\tau=t_j}^{T} w^{(\tau)} \varphi_j^{(\tau)} \\
&= r_j
\end{aligned}
$$

where the third equality is by Lemma F.4.

- **F4 Equal-Time Desirability.** For all $i, j \in N$ s.t. $i \neq j$, if $t_i = t_j$ and the following condition holds:

$$
\left( \exists B \subseteq N \setminus \{i, j\} \ v_{B \cup \{i\}} > v_{B \cup \{j\}} \right) \wedge \left( \forall C \subseteq N \setminus \{i, j\} \ v_{C \cup \{i\}} \geq v_{C \cup \{j\}} \right),
$$

then by Lemma F.6,

$$
\begin{aligned}
r_i &= \sum_{\tau=0}^{t_i - 1} w^{(\tau)} \varphi_i^{(\tau)} + \sum_{\tau=t_i}^{T} w^{(\tau)} \varphi_i^{(\tau)} \\
&= \sum_{\tau=0}^{t_i - 1} w^{(\tau)} v_i + \sum_{\tau=t_i}^{T} w^{(\tau)} \varphi_i \left( v_{(\cdot)}^{(\tau)}, N_\tau \right) \\
&> \sum_{\tau=0}^{t_j - 1} w^{(\tau)} v_j + \sum_{\tau=t_j}^{T} w^{(\tau)} \varphi_j \left( v_{(\cdot)}^{(\tau)}, N_\tau \right) \\
&= r_j \ .
\end{aligned}
$$

- **F5 Uselessness.** By definition $v_i = 0$ if $i$ is *useless*. Following Lemma F.5, $\forall i \in N$, if $\forall C \subseteq N \setminus \{i\}$ $v_{C \cup \{i\}} = v_C$,

$$
r_i = \sum_{\tau=0}^{t_i - 1} w^{(\tau)} \underbrace{v_i}_{0} + \sum_{\tau=t_i}^{T} w^{(\tau)} \underbrace{\varphi_i \left( v_{(\cdot)}^{(\tau)}, N_\tau \right)}_{0} = 0 \ .
$$

- **F6 Necessity.** If both $i$ and $j$ are *necessary*, then $v_i = v_j = 0$, and no coalition can generate value until both of them join the collaboration, which results in the Shapley values of $0$ for all parties. Also, Lemma F.3 guarantees that the Shapley values of $i$ and $j$ are always equal after both of them join. Let $t = \max(t_i, t_j)$, then

$$
\begin{aligned}
r_i &= \sum_{\tau=0}^{t - 1} w^{(\tau)} \varphi_i^{(\tau)} + \sum_{\tau=t}^{T} w^{(\tau)} \varphi_i^{(\tau)} \\
&= \sum_{\tau=0}^{t - 1} w^{(\tau)} \cdot 0 + \sum_{\tau=t}^{T} w^{(\tau)} \varphi_i \left( v_{(\cdot)}^{(\tau)}, N_\tau \right) \\
&= \sum_{\tau=0}^{t - 1} w^{(\tau)} \varphi_j^{(\tau)} + \sum_{\tau=t}^{T} w^{(\tau)} \varphi_j \left( v_{(\cdot)}^{(\tau)}, N_\tau \right) \\
&= r_j \ .
\end{aligned}
$$

- **F7 Time-based Monotonicity.** Suppose $t'$ is the new time value vector such that $t'_i < t_i$ and $t'_j = t_j \ \forall j \neq i$. We divide the entire collaboration duration into three segments, and let $\varphi_i^{(\tau)}, \varphi_i^{(\tau)'}$ be defined based on $t$ and $t'$, respectively. Then,

$$
\begin{aligned}
r_i &= \sum_{\tau=0}^{t'_i - 1} w^{(\tau)} \varphi_i^{(\tau)} + \sum_{\tau=t'_i}^{t_i - 1} w^{(\tau)} \varphi_i^{(\tau)} + \sum_{\tau=t_i}^{T} w^{(\tau)} \varphi_i^{(\tau)} \\
&= \sum_{\tau=0}^{t'_i - 1} w^{(\tau)} \varphi_i^{(\tau)'} + \sum_{\tau=t'_i}^{t_i - 1} w^{(\tau)} v_i + \sum_{\tau=t_i}^{T} w^{(\tau)} \varphi_i^{(\tau)'} \\
&\leq \sum_{\tau=0}^{t'_i - 1} w^{(\tau)} \varphi_i^{(\tau)'} + \sum_{\tau=t'_i}^{t_i - 1} w^{(\tau)} \varphi_i^{(\tau)'} + \sum_{\tau=t_i}^{T} w^{(\tau)} \varphi_i^{(\tau)'} \\
&= r'_i
\end{aligned}
$$

where the inequality is due to Lemma F.1 since party $i$ has joined the collaboration based on $t'$ but not $t$ for $t'_i \leq \tau < t_i$.

- **F8 Time-based Strict Monotonicity.** By $\mathbb{I}_i$, $\exists C \subseteq \{j : t_j < t_i\}$ such that $v_{C \cup \{i\}} > v_C + v_i$. Thus, there exists $t'_i \leq \hat{\tau} < t_i$ such that $\exists \hat{C} \subseteq N_{\hat{\tau}}$ with $v_{\hat{C} \cup \{i\}} > v_{\hat{C}} + v_i$. By a similar argument in Lemma F.1, $\varphi_i^{(\hat{\tau})'} = \varphi_i(v_{(\cdot)}^{\hat{\tau}}, N_{\hat{\tau}}) > v_i = \varphi_i^{(\hat{\tau})}$. Also note that $\varphi_i^{(\tau)'} \geq \varphi_i^{(\tau)} \ \forall \tau$ from the proof of F7. Therefore,

$$
r_i = w^{(\hat{\tau})} \varphi_i^{(\hat{\tau})} + \sum_{\tau \neq \hat{\tau}} w^{(\tau)} \varphi_i^{(\tau)} < w^{(\hat{\tau})} \varphi_i^{(\hat{\tau})'} + \sum_{\tau \neq \hat{\tau}} w^{(\tau)} \varphi_i^{(\tau)'} = r'_i \ .
$$

## F.2 Discussion on [35]

[35] has shown that $v_{(\cdot, \boldsymbol{\lambda})}$ (defined in Sec. 6.2) inherits the superadditivity from $v$, i.e., if $v$ is superadditive, then $v_{(\cdot, \boldsymbol{\lambda})}$ is also superadditive. This is useful because if $v_{(\cdot, t)}$ (defined in Sec. 6.2) inherits the desirable properties of $v$ (e.g., A1, A2, A3 in Sec. 5), it can be shown that the preconditions of Lemma F.1–F.6 are fulfilled, allowing us to use these lemmas to prove Theorem 6.3 in App. F.3. Indeed, we additionally show the inheritance of non-negativity (A1) and monotonicity (A2), in addition to superaddivity (A3), as outlined by the following propsition:

**Proposition F.8** ([35]). *Given a valuation function for the aggregated data of a coalition $v_{(\cdot)} : 2^N \to \mathbb{R}$, and joining time values $t \in \mathbb{Z}_{\geq 0}^n$, the time-aware valuation function $v_{(\cdot, t)} : 2^N \times \mathbb{Z}_{\geq 0}^n \to \mathbb{R}$ as defined in (5) satisfies the following:*

- *If $v_{(\cdot)}$ is non-negative, then $v_{(\cdot, t)}$ is non-negative.*

- *If $v_{(\cdot)}$ is monotonic, then $v_{(\cdot, t)}$ is monotonic.*

- *If $v_{(\cdot)}$ is superadditive, then $v_{(\cdot, t)}$ is superadditive.*

*Proof.* The inheritance of superadditivity follows directly from Proposition 4.1 in [35]. The preservation of non-negativity and monotonicity can be shown similarly following the proof of Proposition 4.1:

Following the definitions in [35], we define a list of increasing values $y_h$, for $h = 0, 1, \ldots, r$ where $r$ is the number of distinct cooperation abilities in $\{\lambda_i\}_{i \in N}$. Specifically, $y_0 = 0$, $y_h = \min_{i \in N} \{\lambda_i : \lambda_i > y_{h-1}\}$. We can interpret $(y_h)_{h=0,1,\ldots,r}$ based on the order of arrival, i.e., $y_h$ is arranged in increasing order of cooperative ability (parties arriving earlier have higher cooperative ability). We also denote the valuation function restricted to a coalition $S \subseteq N$ as $v_{|S}(T) \triangleq v(T \cap S)$ for all $T \subseteq N$. It is shown in [35] that

$$
v_{(\cdot, \boldsymbol{\lambda})} = \sum_{h=0}^{r-1} (y_{h+1} - y_h) \, v_{|N_{h+1}} + \sum_{i \in N} (1 - \lambda_i) \, v_{|\{i\}} \tag{7}
$$

where $N_{h+1} = \{i \in N : \lambda_i \geq y_{h+1}\}$, for $h = 0, \ldots, r - 1$. $N_{h+1}$ represents the set of parties with cooperative ability above a certain threshold (arriving earlier than a specific time). By noting that

if $v$ is non-negative (monotonic), then $v_{|S}$ is non-negative (monotonic) $\forall S \subseteq N$, the inheritance of non-negativity and monotonicity by $v_{(\cdot,t)}$ follows from (7) and $\lambda_i = e^{-\gamma t_i} \in (0,1]$ $\forall i \in N$. $\qquad \square$

### F.3 Proof of Theorem 6.3

- **F1 Non-negativity.** By Proposition F.8, $v_{(\cdot,t)}$ is non-negative and superadditive, then by Corollary F.2, for all $i \in N$,

$$r_i = \varphi_i(v_{(\cdot,t)}, N) \geq 0 \,.$$

- **F2 Individual Rationality.** By Proposition F.8, $v_{(\cdot,t)}$ is superadditive, then by Lemma F.1, $\forall i \in N$,

$$r_i = \varphi_i(v_{(\cdot,t)}, N) \geq v_{(\{i\},t)} = d(v, \{i\}) = v_i \,.$$

- **F3 Equal-Time Symmetry.** For all $i, j \in N$ s.t. $i \neq j$, if $t_i = t_j$ and $\forall C \subseteq N \setminus \{i,j\}$ $v_{C \cup \{i\}} = v_{C \cup \{j\}}$, then $\lambda_i = \lambda_j$. Also, by (7), for all $C \subseteq N \setminus \{i,j\}$

$$
\begin{aligned}
v_{(C \cup \{i\},t)} &= \sum_{h=0}^{r-1} (y_{h+1} - y_h)\, v_{|N_{h+1}}(C \cup \{i\}) + \sum_{k \in N} (1 - \lambda_k)\, v_{|\{k\}}(C \cup \{i\}) \\
&= \sum_{h=0}^{r-1} (y_{h+1} - y_h)\, v_{|N_{h+1}}(C \cup \{j\}) + \sum_{k \in N \setminus \{i,j\}} (1 - \lambda_k)\, v_{|\{k\}}(C \cup \{i\}) \\
&\quad + (1 - \lambda_i) \underbrace{v_{|\{i\}}(C \cup \{i\})}_{v_i} + (1 - \lambda_j) \underbrace{v_{|\{j\}}(C \cup \{i\})}_{v_\emptyset} \\
&= \sum_{h=0}^{r-1} (y_{h+1} - y_h)\, v_{|N_{h+1}}(C \cup \{j\}) + \sum_{k \in N \setminus \{i,j\}} (1 - \lambda_k)\, v(C \cap \{k\}) \\
&\quad + (1 - \lambda_j)\, v_j + (1 - \lambda_i) \cdot 0 \\
&= \sum_{h=0}^{r-1} (y_{h+1} - y_h)\, v_{|N_{h+1}}(C \cup \{j\}) + \sum_{k \in N \setminus \{i,j\}} (1 - \lambda_k)\, v_{|\{k\}}(C \cup \{j\}) \\
&\quad + (1 - \lambda_j)\, v_{|\{j\}}(C \cup \{j\}) + (1 - \lambda_i)\, v_{|\{i\}}(C \cup \{j\}) \\
&= \sum_{h=0}^{r-1} (y_{h+1} - y_h)\, v_{|N_{h+1}}(C \cup \{j\}) + \sum_{k \in N} (1 - \lambda_k)\, v_{|\{k\}}(C \cup \{j\}) \\
&= v_{(C \cup \{j\},t)} \,.
\end{aligned}
$$

Hence, we can use Lemma F.4 as the preconditions are met,

$$r_i = \varphi_i(v_{(\cdot,t)}, N) = \varphi_j(v_{(\cdot,t)}, N) = r_j \,.$$

- **F4 Equal-Time Desirability.** First note that if $\forall C \subseteq N \setminus \{i,j\}$ $v_{C \cup \{i\}} \geq v_{C \cup \{j\}}$, then we have $v_{|S}(C \cup \{i\}) \geq v_{|S}(C \cup \{j\})$ $\forall S \subseteq N$. By (7), since $t_i = t_j$ implies $\lambda_i = \lambda_j$, we have $v_{C \cup \{i\},t} \geq v_{C \cup \{j\},t}$.

Following the notations in App. F.2, $N_1 = N$ since $\lambda_i = \exp(-\gamma t_i) > 0 = y_0 \ \forall i \in N$. Hence, by (7), if $t_i = t_j$ and $\exists B \subseteq N \setminus \{i, j\} \ v_{B \cup \{i\}} > v_{B \cup \{j\}}$, then

$$
\begin{aligned}
v_{(B \cup \{i\}, \boldsymbol{t})} &= \sum_{h=0}^{r-1} (y_{h+1} - y_h) \, v_{|N_{h+1}}(B \cup \{i\}) + \sum_{k \in N} (1 - \lambda_k) \, v_{|\{k\}}(B \cup \{i\}) \\
&\geq \sum_{h=1}^{r-1} (y_{h+1} - y_h) \, v_{|N_{h+1}}(B \cup \{i\}) + \sum_{k \in N} (1 - \lambda_k) \, v_{|\{k\}}(B \cup \{j\}) \\
&\quad + (y_1 - y_0) \underbrace{v_{|N_1}(B \cup \{i\})}_{v_{B \cup \{i\}}} \\
&> \sum_{h=1}^{r-1} (y_{h+1} - y_h) \, v_{|N_{h+1}}(B \cup \{j\}) + \sum_{k \in N} (1 - \lambda_k) \, v_{|\{k\}}(B \cup \{j\}) \\
&\quad + (y_1 - y_0) \underbrace{v_{|N_1}(B \cup \{j\})}_{v_{B \cup \{j\}}} \\
&= v_{(B \cup \{j\}, \boldsymbol{t})} \ .
\end{aligned}
$$

Thus, we have shown that the conditions in Lemma F.6 also hold for $v_{(\cdot, \boldsymbol{t})}$, so

$$
r_i = \varphi_i(v_{(\cdot, \boldsymbol{t})}, N) > \varphi_j(v_{(\cdot, \boldsymbol{t})}, N) = r_j \ .
$$

- **F5 Uselessness.** First note that if $\forall C \subseteq N \setminus \{i\} \ v_{C \cup \{i\}} = v_C$, then $v_{|S}(C \cup \{i\}) = v_{|S}(C) \ \forall S \subseteq N$, which implies $v_{C \cup \{i\}, \boldsymbol{t}} = v_{C, \boldsymbol{t}}$ by (7). Hence, by Lemma F.5, for all $i \in N$, if $\forall C \subseteq N \setminus \{i\} \ v_{C \cup \{i\}} = v_C$,
$$
r_i = \varphi_i(v_{(\cdot, \boldsymbol{t})}, N) = 0 \ .
$$

- **F6 Necessity.** By Proposition 5.1 in [35], $\forall C \subseteq N$ if $\{i, j\} \not\subseteq C \implies v_C = 0$, then
$$
r_i = \varphi_i(v_{(\cdot, \boldsymbol{t})}, N) = \varphi_j(v_{(\cdot, \boldsymbol{t})}, N) = r_j \ .
$$

- **F7 Time-based Monotonicity.** If $(t_i' < t_i) \wedge (\forall j \in N \setminus \{i\} \ t_j' = t_j)$, then by Proposition 4.2 in [35] and the superadditivity of $v$,
$$
r_i' = \varphi_i(v_{(\cdot, \boldsymbol{t}')}, N) \geq \varphi_i(v_{(\cdot, \boldsymbol{t})}, N) = r_i \ .
$$

- **F8 Time-based Strict Monotonicity.** The proof of time-based strict monotonicity follows closely from the proof of Proposition 4.2 in [35] except that we show under $\mathbb{I}_i$, the equality can be removed to achieve strict monotonicity. We only outline the significant changes and refer the readers to [35] for more details.

  **Notations.** We follow the notations defined in App. F.2. Additionally, we define the zero-normalized valuation function $v_o$ as $v_o(S) \triangleq v(S) - \sum_{k \in S} v(k) \ \forall S \subseteq N$. Also recall that in time-aware CML setting, $\lambda_i = \exp(-\gamma t_i)$.

  **Proof Sketch of [35].** [35] showed that if $t_i' < t_i \ (\lambda_i' > \lambda_i)$ and $t_j' = t_j \ \forall j \neq i \ (\lambda_j' = \lambda_j \ \forall j \neq i)$, then the difference $\varphi_i(v_{(\cdot, \boldsymbol{t}')}, N) - \varphi_i(v_{(\cdot, \boldsymbol{t})}, N)$ can be expressed as linear combinations of the Shapley values, where the Shapley values are calculated with different valuation functions under separate conditions. There are a total of six possibilities.[10] Specifically:

  - Under Condition 3,
  $$
  \varphi_i(v_{(\cdot, \boldsymbol{t}')}, N) - \varphi_i(v_{(\cdot, \boldsymbol{t})}, N) = \eta_3 \varphi_i(v_{o|C_3}, N) + \eta_3' \varphi_i(v_{o|C_3'}, N) \ ;
  $$

  - Under Condition 6,
  $$
  \varphi_i(v_{(\cdot, \boldsymbol{t}')}, N) - \varphi_i(v_{(\cdot, \boldsymbol{t})}, N) = \eta_6 \varphi_i(v_{o|C_6}, N) + \eta_6' \varphi_i(v_{o|C_6'}, N) \ ;
  $$

  - Under Condition $\alpha$ for $\alpha \in \{1, 2, 4, 5\}$,
  $$
  \varphi_i(v_{(\cdot, \boldsymbol{t}')}, N) - \varphi_i(v_{(\cdot, \boldsymbol{t})}, N) = \eta_\alpha \varphi_i(v_{o|C_\alpha}, N) \ .
  $$

---

[10]To be more precise, there is one more possibility whose proof is omitted in [35], but this possibility is covered in F7. The readers can refer to the footnote of F7 in Sec. 4 for details.

Here, $\eta_\alpha, \eta_3', \eta_6' > 0$ are positive coefficients and $C_\alpha, C_3', C_6' \subseteq N$ are coalitions for all $\alpha \in \{1, 2, 3, 4, 5, 6\}$. We refer the readers to [35] for actual values of $\eta_\alpha, C_\alpha$, as well as the specifics about the conditions. The monotonicity then follows from the non-negativity of the Shapley values.

**Strict Monotonicity.** To enforce strict inequality, we need to show

$$\varphi_i(v_{o|C_\alpha}, N) > 0 \ \forall \alpha \in \{1, 2, 4, 5\} \tag{8}$$

and

$$\varphi_i(v_{o|C_\alpha}, N) > 0 \vee \varphi_i(v_{o|C_\alpha'}, N) > 0 \ \forall \alpha \in \{3, 6\} . \tag{9}$$

Let $T = \{i : t_j < t_i\}$ and $T' = T \cup \{i\}$. It is easy to check that $T' \subseteq C_\alpha \ \forall \alpha \in \{1, 2, 4, 5\}$ and $T' \subseteq C_\alpha \vee T' \subseteq C_\alpha' \ \forall \alpha \in \{3, 6\}$. This relieves our burden of checking every single possible condition, as we can restrict ourselves to only consider $S \subseteq T' \setminus \{i\}$. In fact, To show both (8) and (9) hold (positive Shapley value), it is sufficient to show $v_{o|T'}(S \cup \{i\}) - v_{o|T'}(S) > 0$ for some $S \subseteq T' \setminus \{i\}$, i.e.,

$$\exists S \subseteq T' \setminus \{i\} \ \ v_o((S \cup \{i\}) \cap T') - v_o(S \cap T') > 0 . \tag{10}$$

Since $(S \cup \{i\}) \cap T' = (S \cap T') \cup (\{i\} \cap T') = (S \cap T') \cup \{i\}$, by the inherited superadditivity of $v_o$, as long as $\exists S \subseteq T' \setminus \{i\}$ such that $v_o((S \cap T') \cup \{i\}) > v_o(S \cap T') + v_o(\{i\})$, (10) is satisfied.

By the definition of zero-normalization, $v_o(\{i\}) = 0$, so we need $v_o((S \cap T') \cup \{i\}) > v_o(S \cap T')$. Again, by the definition of $v_o$, we require

$$v((S \cap T') \cup \{i\}) - \sum_{k \in (S \cap T') \cup \{i\}} v(\{k\}) > v(S \cap T') - \sum_{k \in S \cap T'} v(\{k\}) . \tag{11}$$

Since $i \notin S$, (11) is equivalent to $v((S \cap T') \cup \{i\}) - v(\{i\}) > v(S \cap T')$. Let $B \subseteq T = T' \setminus \{i\}$ be the coalition that satisfies $\mathbb{I}_i$. Then, if we take $S = B$, we have $S \cap T' = B$ because $S \subseteq T'$. Hence, $v((S \cap T') \cup \{i\}) - v(\{i\}) = v_{B \cup \{i\}} - v_i > v_B$ by $\mathbb{I}_i$. Therefore, (11) is also satisfied so both (8) and (9) are true and

$$r_i' - r_i = \varphi_i(v_{(\cdot, \boldsymbol{t'})}, N) - \varphi_i(v_{(\cdot, \boldsymbol{t})}, N) > 0 \implies r_i' > r_i .$$

### F.4  Efficient Computation of Equation (5)

We can skip computing the Harsanyi dividend [14] by using Eq. 7 in App. F.2. In the following, we explain how to implement this efficient calculation in practice and provide insight into this approach.

We obtain $\acute{e}$ by sorting the weights $(e^{-\gamma t_j})_{j \in C}$ in descending order and appending a 0. We obtain $\acute{C}$ by sorting the list of parties in coalition $C$ in ascending order of their joining time values. Then,

$$v_{C, \boldsymbol{t}} = \sum_{j=1}^{|C|} v_{\acute{C}_{[:j]}}[\acute{e}_{[j]} - \acute{e}_{[j+1]}] + \sum_{j=1}^{|C|} [1 - \acute{e}_{[j]}] v_{\acute{C}_{[j]}} \tag{12}$$

only involves summing $|C|$ terms.[11]

We observe that for the $k$th ranked party $i = \acute{C}_{[k]}$, the dividend $d(v, \{i\})$ contributes to any $v_C$ where $i \in C$. Thus, the dividend has a weight of $\sum_{j=k}^{|C|} [\acute{e}_{[j]} - \acute{e}_{[j+1]}] + [1 - \acute{e}_{[k]}] = 1$ in Eq. (12) which is the same as in Eq. (5).

In addition, the dividend $d(v, T)$ where $|T| \geq 2$ contributes to $v_C$ only if $T \subseteq C$. Let $k$ be the rank of the latest party in $T$. The dividend would have a weight of $\sum_{j=k}^{|C|} [\acute{e}_{[j]} - \acute{e}_{[j+1]}] = \acute{e}_{[k]} = \min_{j \in T} \boldsymbol{e}_{[j]}$ in Eq. (12) which is the same as in Eq. (5).

## G  Reward Realization Methods

**Likelihood Tempering.** One way to realize $(r_i^*)_{i \in N}$ *exactly* when using conditional IG (2) for data valuation is the *likelihood tempering* [51] method. The mediator assigns party $i$ a model reward

---

[11] $[j]$ and $[:j]$ denote indexing the $j$-th element and up to and inclusive of the $j$-th element.

(posterior) $p_i^*$ that is updated by its likelihood $p(D_i|\boldsymbol{\theta})$ and the tempered likelihood of others' data $\propto p(D_{N\setminus\{i\}}|\boldsymbol{\theta})^{\kappa_i}$ where the tempering factor $\kappa_i \in [0,1]$. Note that when $\kappa_i = 0$ and 1, party $i$'s model is trained on $D_i$ and $D_N$, respectively. We can view $D_{N\setminus\{i\}}$ as partitioned into two random variables $R_i$ and $R_{-i}$ with likelihood proportional to $p(D_{N\setminus\{i\}}|\boldsymbol{\theta})^{\kappa_i}$ and $p(D_{N\setminus\{i\}}|\boldsymbol{\theta})^{1-\kappa_i}$, respectively. Note that as $D_{N\setminus\{i\}}$ follows a Gaussian distribution, $R_i$ and $R_{-i}$ also follow Gaussian distributions (but have larger variances). The mediator then uses any root-finding algorithm to exactly solve for $\kappa_i$ such that $I(\boldsymbol{\theta}; D_i \cup R_i | R_{-i}) = r_i^*$.

**Subset Selection.** The mediator can generate model rewards of different values for each party $i \in N$ by training on different subsets of $D_N$. Since $r_i^*$ exceeds $v_i$ (i.e., F2 is satisfied), party $i$ receives a model reward trained on $D_i^r \triangleq D_i \cup R_i$ where $R_i \subseteq D_{N\setminus\{i\}}$ is selected such that $v(D_i^r) = r_i^*$. However, as it is intractable to enumerate an exponential number of subsets to find $D_i^r$ whose value $v(D_i^r)$ is the closest to $r_i^*$, we resort to an approximation method. For example, when conditional IG (2) is used as the valuation function, we shuffle $D_{N\setminus\{i\}}$ and incrementally add points to $R_i$ only until $I(\boldsymbol{\theta}; D_i^r | D_N \setminus D_i^r) > r_i^*$.

# H   Additional Experiment Details and Results

All experiments were performed on a system equipped with an NVIDIA A16 GPU with 10 GB of VRAM. The system was configured with NVIDIA driver version 515.43.04 and CUDA version 11.7.

## H.1   Experiment Setup

Part of our experiment setup is adapted from [49]. We empirically verify our results on an additional *diabetes progression* (DiaP) dataset [9] with conditional IG as the valuation function. We also demonstrate our results on the CIFAR-100 dataset [23] with $n = 10$ parties, and the dual of validation accuracy is used as the valuation function. We use the Gaussian Process (GP) regression model for all datasets except MNIST and CIFAR-100, for which we train neural networks (NNs). For all GP models, we use *automatic relevance determination* such that each input feature has a different lengthscale parameter. For the Friedman, CaliH and DiaP datasets, we use an 80–20 train-test split to obtain $D_{\text{train}}$ and $D_{\text{test}}$, all parties' data are randomly sampled without replacement from $D_{\text{train}}$.

**Synthetic Friedman Dataset ($n_1 = n_2 > n_3$)** We generate data based on the Friedman function:

$$y = 10\sin(\pi\boldsymbol{x}_{[d,0]}\boldsymbol{x}_{[d,1]}) + 20(\boldsymbol{x}_{[d,2]} - 0.5)^2 + 10\boldsymbol{x}_{[d,3]} + 5\boldsymbol{x}_{[d,4]} + 0\boldsymbol{x}_{[d,5]} + \mathcal{N}(0,1)$$

where $d$ is the index of data point $(\boldsymbol{x}, y)$ and its 6 input features (i.e., $\boldsymbol{x} \triangleq (\boldsymbol{x}_{[d,a]})_{a=0,\dots,5}$) are independent and uniformly distributed over the input domain $[0,1]$. The last input feature $\boldsymbol{x}_{[d,5]}$ is an independent variable which does not affect the output $y$.

We standardize the values of output $y$ and train a GP model with a squared exponential kernel. We consider a test set with 200 points and parties 1, 2 and 3 having $n_1 = 300$, $n_2 = 300$ and $n_3 = 200$ training points, respectively.

**CaliH Dataset ($n_1 > n_2 > n_3$)** We standardize each input feature $\boldsymbol{X}$ and the output vector $\boldsymbol{y}$ to have a variance of 1 and train a GP model with a squared exponential kernel.

We consider a test set with 4128 points and parties 1, 2 and 3 having $n_1 = 600$, $n_2 = 400$ and $n_3 = 200$ training points, respectively.

**DiaP Dataset ($n_1 = n_2 < n_3$)** We use the diabetes progression (DiaP) dataset [9] with scaled features from sklearn. We remove the gender feature and standardize the output vector $\boldsymbol{y}$ to have a variance of 1.

We train a GP model with a composite kernel comprising the squared exponential kernel and the exponential kernel.

We use an 80-20 train-test split to obtain a test set with 88 points and parties 1, 2 and 3 having $n_1 = 75$, $n_2 = 75$ and $n_3 = 125$ training points, respectively.

**MNIST Dataset** Since NNs are highly effective on the MNIST dataset, we first create a subsampled version by randomly selecting 20000 data points from the original training set, which serves as the aggregated data $D_{\text{train}}$ for all parties. $D_{\text{train}}$ is then distributed among the three parties based on data point labels. Specifically, the 10 classes are randomly divided into three

groups of size 3, 3 and 4. Party $i$ receives all the data points in $D_{\text{train}}$ with labels in the $i$th group.

We train a NN with one hidden layer of 16 neurons using ReLU activation functions and stochastic gradient descent. The model with the highest validation accuracy over 10 training epochs is selected.

**CIFAR-100 Dataset** We use all 50,000 training images as the aggregated dataset for all parties. Prior to data assignment, we preprocess the images using a pretrained ResNet50 backbone to extract embedding features. Each of the 10 parties is then assigned all the transformed data from 10 randomly selected classes. There is no data overlap between parties, and collectively they cover the entire set of 50,000 training images.

We train a NN with two hidden layers containing 1024 and 512 neurons each. We use ReLU as the activation function, and the NN is trained using dropout and the Adam optimizer. The model with the highest validation accuracy over 20 training epochs is selected.

## H.2 Metrics

**Information Gain** Let $X$ be the input matrix and $y$ is the output vector. In a GP model with function $f$, the latent output vector is $f \triangleq f(X)$. We assume that $y$ is generated from $f$ by adding independent Gaussian noise with noise variance $\sigma^2$. The information gain of the GP from evaluating at $X$ is given by

$$\mathbb{I}(f; X) = \mathbb{I}(f; X) = 0.5 \log(|I + K_{XX}/\sigma^2|) \tag{13}$$

where $K_{XX}$ is a covariance matrix with components $k(x, x')$ for all $x, x'$ in $X$ and $k$ is a kernel function.

When calculating the IG for heteroscedastic data (i.e., data points with different noise variances), instead of dividing by $\sigma^2$, we multiply by the diagonal matrix $K_{\text{noise}}^{-1}$ such that each diagonal component of $K_{\text{noise}}$ represents the noise variance corresponding to a data point in $D$.

**Mean Negative Log Probability** The model reward $p_i^*$ is Gaussian process posterior after observing the dataset $D_i$ (and some subset of $D_{N \setminus i}$). To compute the MNLP, we compute the model reward $p_i^*$'s (predictive) mean $\mu_x$ and variance $\sigma_x^2$ at each test point $x$. Then,

$$\text{MNLP} = \frac{1}{|D_{\text{test}}|} \sum_{(x,y) \in D_{\text{test}}} \frac{1}{2} \left( \log(2\pi\sigma_x^2) + \frac{(\mu_x - y)^2}{\sigma_x^2} \right). \tag{14}$$

A lower MNLP is better as it indicates that the model is more confident in its prediction (small first term) and is not overconfident in predictions with larger squared error (small second term).

## H.3 Reward Values on the DiaP Dataset

In Fig. 8, we create a scenario where $v_1 = 36.35$ is close to $v_2 = 37.11$, while $v_3 = 61.79$ is significantly larger. When $t_1 = t_2 = t_3 = 0$, the Shapley values are $\varphi_1 = 38.69, \varphi_2 = 39.46, \varphi_3 = 64.53$, with party 3 receiving model with the highest possible value $v_N$. As party 1 joins later ($t_1$ increases), Fig. 8 shows that its reward $r_1^*$ decreases as a disincentive for joining late. While F8 only stipulates a decrease in $r_1^*$, we also see a drop in $r_2^*$ and $r_3^*$. This is because other parties receive benefits from party 1's collaboration for a shorter duration, resulting in lower total rewards. However, each party is still guaranteed individual rationality (F2), i.e., all parties receive rewards at least as valuable as their own data (plotted as grey horizontal lines in Fig. 8).

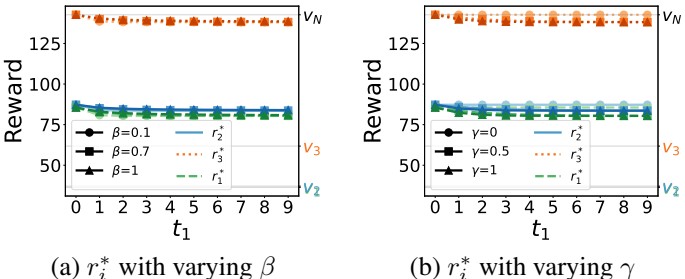

(a) $r_i^*$ with varying $\beta$        (b) $r_i^*$ with varying $\gamma$

Figure 8: Graphs of reward values $r_i^*$ vs. $t_1$ with the DiaP dataset using (a) time-aware reward cumulation and (b) time-aware data valuation.

The trend in Fig. 8 is nearly horizontal and thus almost unnoticeable. This is because the changes in reward values are small compared to the largest reward value $v_N$. To illustrate the trend more clearly, we restrict the range of reward values. In Figs. 9a-b, we can clearly observe that the reward values of parties 1 and 2 decrease with the joining time $t_1$ when the reward values are restricted to the range $[80, 90]$. Similarly, the reward value of party 3 also decreases, as shown in Figs. 9c-d, when the range is restricted to $[135, 145]$.

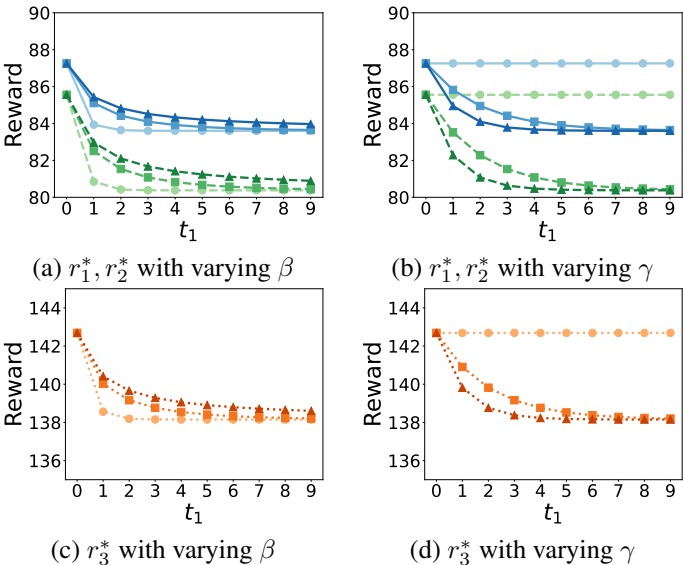

(a) $r_1^*, r_2^*$ with varying $\beta$      (b) $r_1^*, r_2^*$ with varying $\gamma$

(c) $r_3^*$ with varying $\beta$       (d) $r_3^*$ with varying $\gamma$

Figure 9: Graphs of reward values $r_1^*$ (green), $r_2^*$ (blue), $r_3^*$ (orange) vs. $t_1$ using (a,c) time-aware reward cumulation and (b,d) time-aware data valuation with the DiaP dataset.

### H.4 Reward Realization with Likelihood Tempering Method on the DiaP Dataset

We verify that when using likelihood tempering, models with higher reward values $r_i^*$ also have better predictive performance (lower MNLP). Thus, in Fig. 10, we observe that: (i) Each party receives a model reward with lower MNLP than the model it trained alone. (ii) When a party delays its participation, it receives a model with worse MNLP. (iii) Parties with higher Shapley or data values receive models with better MNLP. In addition, using a smaller $\beta$ and a larger $\gamma$ leads to a decrease in $r_1^*$ and an increase in MNLP. We also note that the increasing trend of MNLP is not obvious in Fig. 10. This is because the change in reward values is small, as explained in the App. H.3.

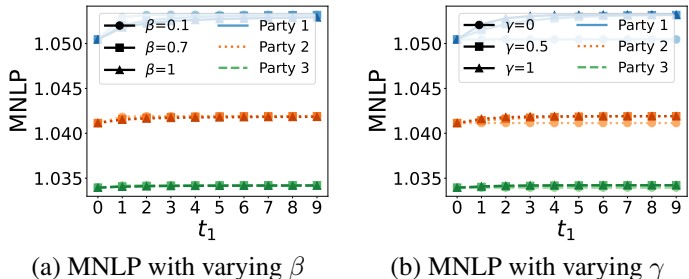

(a) MNLP with varying $\beta$      (b) MNLP with varying $\gamma$

Figure 10: Graphs of MNLP vs. $t_1$ using (a) time-aware reward cumulation and (b) time-aware data valuation using likelihood tempering for reward realization on the DiaP dataset.

## H.5 Reward Realization with Subset Selection Method on the DiaP Dataset

When using subset selection, we similarly observe that models with higher reward values $r_i$ have better or similar predictive performance. Based on Fig. 11, we conclude that: (i) Each party receives a model reward with lower MNLP than the model it trained alone. (ii) When a party delays its participation, it does not receive a model with better MNLP. (iii) Parties with higher Shapley or data values receive models with better MNLP.

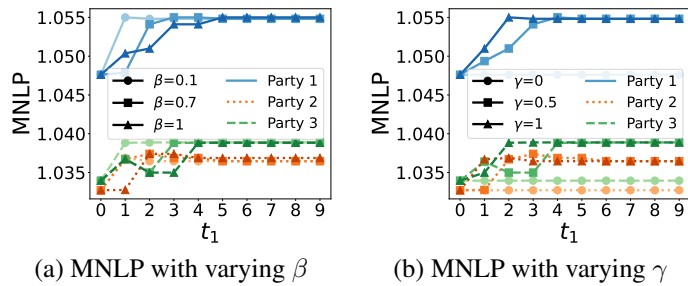

(a) MNLP with varying $\beta$      (b) MNLP with varying $\gamma$

Figure 11: Graphs of MNLP vs. $t_1$ using (a) time-aware reward cumulation and (b) time-aware data valuation using subset selection for reward realization on the DiaP dataset.

In Sec. H.4, we observe that the MNLP increases smoothly as $t_1$ increases. However, for subset selection, the MNLP exhibits less smooth behavior. This is because the reward values can only be realized approximately, and the process is sensitive to the quality of the data points included in the subset. For example, the MNLP would increase by a larger extent with the addition of an outlier or the removal of all data from a region of the input space.[12]

## H.6 Reward Values with 10 Parties on the CIFAR-100 Dataset

As in Sec. 7, we investigate how each party's reward changes with the joining time of Party 1. We set $\beta = \gamma = 1$ for both methods. Fig. 12 shows that Party 1's reward decreases with later joining times under both methods, consistent with our theoretical guarantees and serving as a disincentive for joining late. More importantly, Fig. 12 demonstrates that our approaches remain consistent when an efficient approximation method is adopted, even as the number of parties increases and a larger real-world dataset is used.

---

[12]In contrast, likelihood tempering always use all data and only vary their impact by varying the tempering factor and variance.

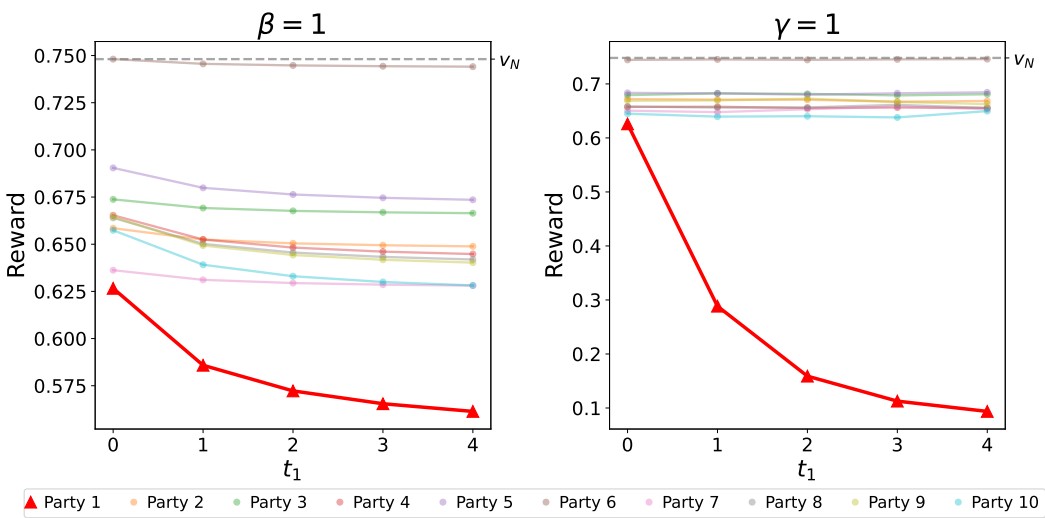

Figure 12: Graphs of reward values $r_i^*$ vs. $t_1$ with the CIFAR-100 dataset using (a) time-aware reward cumulation and (b) time-aware data valuation.

