# OpenReview forum: "Incentivizing Time-Aware Fairness in Data Sharing"
_NeurIPS.cc/2025/Conference — NeurIPS 2025 poster_

### Official Review · Reviewer_XdcU · 2025-06-23

**Clarity:** 2
**Significance:** 2
**Originality:** 2
**Rating:** 4
**Confidence:** 3

**Summary:**

This work considers the problem of heterogeneous arrivals of data contributors. It proposes to provide higher incentives to the ones contributing earlier to motivate participation. The authors proposed two methods to determine the reward values.

**Questions:**

-  How is the data modeled? How does the data distribution affect model quality? This is important because this work emphasizes data sharing rather than game design.
- Why is it reasonable to set higher rewards for earlier participation? I understand the intuition, while a formal mathematical formulation is needed to motivate this setting. That is, there should be an objective with physical meaning, solving which we can obtain that "higher rewards for earlier participation".
- How is fairness modeled in this work? Fairness should be a formal mathematical metric rather than an intuition.

**Ethical Concerns:**

["NO or VERY MINOR ethics concerns only"]

**Final Justification:**

The authors have addressed my comments, and I have a better understanding of the properties and other system settings. To improve this work, I believe it is beneficial to explicitly model the non-IID data   (although it was modelled implicitly using the "conditional information gain"  and can be addressed empirically). It is still more promising to model and analyze the party arrival behavior (rather than assuming their behavior using a reward function). Overall, this work has made theoretical contributions in incentivizing sharing (considering asynchronous arrival behaviors) and provided interesting insights.

**Limitations:**

Yes

**Quality:**

3

**Strengths And Weaknesses:**

Strengths:
- The heterogeneous time arrival problem is interesting.
- The analysis of this work is complete.

Weakness:
- The data was not actually characterized in this work. Specifically, whether non-IID or IID data is considered? If non-IID, then how do the data distributions of different parties affect the reward? If IID, then only the data quantity matters, i.e., no need to distinguish between parties. Further, if the data distribution does not matter, then it basically means the modeling is not related to the data themselves. In this case, the proposed methods should be able to apply to other scenarios without modifications. This is not good as the main feature of data sharing was not characterized. The modeling is too general.
- F1 and F2 are basically the same concept. People usually use individual rationality without mentioning non-negativity.
- In F5, "improving model quality" is non-rigorous. As mentioned earlier, the data together with the model was not characterized in the formulation.
- F7 and F8 are human-invented properties. A more formal way should be formulating a problem and then showing that earlier participation should impose a higher reward. However, in this work, the authors impose the setting that earlier participation should have a higher reward. This may not be optimal sometimes.
- The approach proposed in this work is relatively standard. Again, it is not actually related to data sharing, because those key features were not characterized in the formulation. The proposed approaches are simply existing approaches in coalition games without much modification associated with data sharing.
- How is gamma tuned?
- The approach should be compared with existing works. Although there may not be works considered similar settings, some incentive works should be compared with to show the enhancement for addressing heterogeneous arrivals.
- Fairness is not modeled in this work.

---

> ### Author Rebuttal · Authors · 2025-07-31
>
> We sincerely thank the reviewer for taking the time to review our work and for recognizing our interesting problem and complete analysis. We address your questions below:
>
> > ...whether non-IID or IID data is considered?
> >
> > How is the data modeled? How does the data distribution affect model quality?
>
> **The data distribution can be either IID or non-IID.** Our work adopts the perspective that the value of the data depends on the learned model. That is, both data quantity and data quality matter. Hence, we consider valuing the data based on the conditional information gain on the model parameters or the dual of the accuracy function, which inherently captures the dependency on models and are applicable to all data distributions. Note that this perspective is shared many data valuation works [13, 25] that also value data by the model performance when trained on specific subsets. **In our experiments, we explicitly include both IID data with different data quantities** (Friedman and CaliH datasets, lines 312--314) **and non-IID data distributions** (MNIST and CIFAR-100 datasets, lines 317--318 and lines 910--914). The experiment results in Section 7 demonstrate that our methods consistently capture the relative value of each party under both settings.
>
> > Fairness is not modeled in this work.
> >
> > How is fairness modeled in this work? Fairness should be a formal mathematical metric rather than an intuition.
>
> In our work, **fairness is modeled through satisfying formal incentive-based criteria** (F1--F5 in Sec. 4). As noted in line 128, our work adopts a well-established view in the literature that fairness can be defined in terms of satisfying certain axiomatic properties. For example, in [47], fairness is defined as satisfying incentives such as symmetry, desirability, uselessness which are inspired by intuition. We explicitly include these incentives or generalizations thereof in our formulation (Sec. 4), and our proposed methods are designed to satisfy them, hence achieving fairness.
>
> In addition, we introduce the novel time-aware incentives (F6--F8) to capture time-aware fairness, which mathematically formalize the perspective that earlier parties should be rewarded accordingly. It is a widely recognized perspective in psychology, economics (lines 54--57) and motivated by practical scenarios (lines 45--53). These incentives are formal mathematical/logical constriants that are inspired by intuition like individual rationality and desirability in the literature [37, 47, 52].
>
> > F7 and F8 are human-invented properties.
> >
> > Why is it reasonable to set higher rewards for earlier participation? I understand the intuition, while a formal mathematical formulation is needed to motivate this setting. That is, there should be an objective with physical meaning, solving which we can obtain that "higher rewards for earlier participation".
>
> We respectfully suggest that a formal mathematical formulation or objective with physical meaning may not be strictly necessary to motivate the incentives and setting considered in our work. Incentives and other properties desirable to humans may not be _naturally_ occurring (i.e., solution to objective with physical meaning) but should be _artificially_ guaranteed to realize benefits through well-designed solutions, as often done in economics and game theory. For example, in the seminal work on Shapley value [46], the author proposed the axioms of symmetry, efficiency (total payoff must be the value of the game) and additivity (when two independent games are combined, their values must be added party by party) and showed that the Shapley value satisfies them. These axioms turn out to be useful in many applications such as in data valuation [13] where we want to attribute model accuracy to individual data points. As another example, fairness (equal rewards for equal contribution; higher reward for contributing more valuable data) [39, 47] does not naturally occur but satisfying it would encourage parties to contribute more.
>
> It is "reasonable" to set higher rewards for earlier participation because it is **desirable** in ML applications (as we have explained in Sec. 1 and App. A). Incentives for earlier participation must be designed and provided precisely because parties will not naturally do so. We do define the incentives formally as math equations in Sec. 4.
>
> > F1 and F2 are basically the same concept. People usually use individual rationality without mentioning non-negativity.
>
> We wish to clarify that F1 (non-negativity) and F2 (individual rationality) are not the same concept as F1 ensures all parties receive non-negative rewards while F2 ensures the rewards are at least as valuable as each party's own data. However, if we assume non-negativity of the valuation function (line 205), then F2 implies F1. We still include F1 as one of the incentives as it is only implied under this extra assumption in a later section (Sec. 5), and **it is convention in the literature** as seen in [47, 52].
>
> > In F5, "improving model quality" is non-rigorous. As mentioned earlier, the data together with the model was not characterized in the formulation.
>
> We wish to clarify that in lines 118--120, we define $v_C$ as the model performance/quality achieved by training on the aggregated dataset $D_C$. Accordingly, in F5 (uselessness), the condition $v_{C \cup \{i\}} = v_C$ implies that $i$'s data **fail to improve the model quality**.
>
> > How is gamma tuned?
>
> We demonstrate the effects of different hyperparameters ($\beta$ and $\gamma$) in our experiments (Sec. 7). In practice, the trusted mediators (e.g., a data sharing platform) can test out different values of $\beta$ and $\gamma$ to guide future decisions when the valuation function is the same (e.g., dual of accuracy).
>
> Concretely, the mediator can start by setting $\beta=\gamma=1$, which is shown to provide a reasonable trade-off between incentivizing early participation and encouraging high-quality data in our experiments (Sec. 7). **For future collaboration**, the mediator can then adjust these values based on observed behavior of parties: decrease $\beta$/increase $\gamma$ if early participation is sparse; increase $\beta$/decrease $\gamma$ if the majority of data received seems rushed and of low-quality. In addition, mediators can share their empirical experiences with parameter tuning (without exposing underlying data) to form best practices over time.
>
> > The approach should be compared with existing works. Although there may not be works considered similar settings, some incentive works should be compared with to show the enhancement for addressing heterogeneous arrivals.
>
> We appreciate the reviewer's understanding that there may not be works tackling similar settings. We would like to clarify here and in our revised paper that **comparisons with classical non-time-aware reward schemes are already included in our experiments** (lines 318--322). Specifically, existing methods that do not account for joining time [41, 47] can be seen as special cases of our methods when $\beta \to \infty$ and $\gamma=0$. The baselines correspond to the horizontal lines in our experiment results (e.g., Fig. 3b and Fig. 7a-b), which suggest that they assign constant rewards regardless of joining time, thereby fail to satisfy incentive F8 (time-based strict monotonicity).
>
> > The approach proposed in this work is relatively standard. Again, it is not actually related to data sharing, because those key features were not characterized in the formulation.
>
> We have **addressed the concern regarding IID/non-IID data and improving model quality** by clarifying that **data is valued relative to the model** in our formulation. If there are remaining concerns, we would appreciate if the reviewer could clarify which specific "key features" of data sharing they believe are missing from our formulation, and how these differ from the key incentives in data sharing outlined in Sec. 4, or the various data distributions (both IID and non-IID) considered in our experiments (Sec.7), so that we can respond more precisely.
>
> We would like to emphasize that **our approach is related to data sharing as it addresses a key underexplored practical problem in data sharing**: When parties share their data at heterogeneous times, how should we fairly reward each party based on both their time of arrival and value of data? A key contribution of our work lies in formulating and posing this problem itself.
>
> We would also like to clarify that we propose two reward schemes that incorporate the joining time information to address this problem. These schemes **generalize the standard coalition game approaches** (e.g., the Shapley value) by embedding the time information in the reward cumulation stage (Sec. 6.1) and the valuation stage (Sec. 6.2). Importantly, both schemes provably satisfy all the proposed incentives, this also poses additional technical challenges.
>
> Thank you for your comments! We hope our clarifications above can improve your opinion of our work.

---

> > ### Comment · Reviewer_XdcU · 2025-08-03
> >
> > Thank the authors for addressing my comments. The incorporation of the conditional information gain appears to be reasonable. In addition, based on the rebuttal response, this work provides interesting insights into fair allocation under heterogeneous arrivals.

---

> > > ### Author Response · Authors · 2025-08-04
> > >
> > > Dear Reviewer, we are glad to hear that our rebuttal addressed your concerns, and we sincerely appreciate your recognition of our interesting insights. If there are any remaining concerns, we would be happy to address them during the discussion period.

---

### Official Review · Reviewer_E1vJ · 2025-07-01

**Clarity:** 3
**Significance:** 3
**Originality:** 4
**Rating:** 4
**Confidence:** 3

**Summary:**

This paper tackles the problem of incentivizing parties in collaborative machine learning who join at different times. Unlike prior work assuming simultaneous participation, the authors introduce a time-aware fairness framework that rewards earlier contributors more, reflecting their higher risk and contribution to collaboration momentum. They propose two reward mechanisms—time-aware reward cumulation and time-aware data valuation—both designed to satisfy fairness and new time-sensitive incentive conditions. It is novel to effectively balance data quality and joining time, encouraging early, valuable participation.

**Questions:**

1. I am curious about how to set the parameters in practice to control the emphasis on joining times and early participation. A user may want to know these parameters to decide whether to submit noisy data early or spend more time cleaning and submit high-quality data later.

2. I understand that it may not be possible to guarantee the strong theoretical properties of the reward values under the online FL setting. However, it would be helpful to see a simple extension of your method to this setting along with empirical results. This is important to confirm the practical applicability of your work. If time constraints prevent experimentation, a discussion of your plans to extend the approach to online FL would be valuable.

**Ethical Concerns:**

["NO or VERY MINOR ethics concerns only"]

**Final Justification:**

My final justification for the recommended score is still 4 (borderline accept). The main thing holding me back is parameters like β or γ, I agree with the plan proposed by the author (introducing a mediator), but I am also a bit concerned about its interpretability in practice. I mean, maybe under a different dataset and a different number of participants, the same number of β may have different effects. But considering this is new work proposing the problem of time-aware, I would prefer to keep 'borderline accept'. Meanwhile, I appreciate the author's time and effort in providing more experiments, so I keep my rate of 'borderline accept'.

**Limitations:**

yes

**Paper Formatting Concerns:**

I do not find major formatting issues.

**Quality:**

3

**Strengths And Weaknesses:**

Strengths
- Introduces a novel time-aware fairness framework for CML, addressing asynchronous participation—an underexplored but realistic setting.
- Proposes two theoretically sound methods with formal guarantees for satisfying fairness and time-aware incentives
- Well-structured with clear motivation, incentive definitions, and illustrative examples

Weaknesses
- The paper has already discussed some limitations, such as computational efficiency and the choice of a non-Federated Learning (non-FL) setting. Regarding computational efficiency, I agree that this issue is not unique to this work and therefore should not be considered a limitation specific to this paper. However, I have some questions about the non-FL setting, which I will raise later.
- While I acknowledge the strong theoretical contributions of this paper, I would like to see more experiments on large-scale real-world datasets if time permits. I will elaborate on this point later.

---

> ### Author Rebuttal · Authors · 2025-07-31
>
> We sincerely thank the reviewer for taking the time to review our work and for recognizing that we are tackling a realistic setting with clear motivation. We address your questions below:
>
> > Q1. I am curious about how to set the parameters in practice to control the emphasis on joining times and early participation. A user may want to know these parameters...
>
> We agree that the parameters should be broadcasted before the collaboration starts. In practice, the trusted mediators (e.g., a data sharing platform) can test out different values of $\beta$ and $\gamma$ to guide future decisions when the valuation function is the same (e.g., dual of accuracy).
>
> Concretely, the mediator can start by setting $\beta=\gamma=1$, which is shown to provide a reasonable trade-off between incentivizing early participation and encouraging high-quality data in our experiments (Sec. 7). **For future collaboration**, the mediator can then adjust these values based on observed behavior of parties: decrease $\beta$/increase $\gamma$ if early participation is sparse; increase $\beta$/decrease $\gamma$ if the majority of data received seems rushed and of low-quality. In addition, mediators can share their empirical experiences with parameter tuning (without exposing underlying data) to form best practices over time.
>
> > Q2. I understand that it may not be possible to guarantee the strong theoretical properties of the reward values under the online FL setting. However, it would be helpful to see a simple extension of your method to this setting along with empirical results.
>
> We appreciate the reviewer's understanding that extending to the FL setting may violate the theoretical properties of our methods (lines 560--563). Notably, in an online FL setting, client contributions must be evaluated **dynamically**, and **reward values must be updated and distributed during each communication round**. This introduces a key practical issue: **once rewards are distributed, it becomes difficult to claw back or revise them to ensure overall fairness** if the reward values change when new clients join (lines 580--582).
>
> Nevertheless, we show how to **adapt our time-aware reward cumulation method (Sec. 6.1) to dynamically update the reward values** of each client during each communication round of the online FL setting. We wish to clarify that this adaptation experiment is primarily intended to **showcase the feasibility of extending our method to other settings**, and the **empirical results may not fully satisfy the theoretical properties established in the main paper**.
>
> **Adapting Time-Aware Reward Cumulation to online FL**
>
> Denote $r_i^{(t)}$ as the reward value of client $i$ at communication round $t$. We update $r_i^{(t)}$ via a running average of Shapley values weighted by the penalizing factor $\beta$ on joining time:
> $$r_i^{(t)} = \frac{r_i^{(t-1)} \cdot t + \phi_i^{(t)} \cdot \beta^t}{t + 1}$$
> where $\phi_i^{(t)}$ is the modified Shapley value defined in line 246. For computing $\phi_i^{(t)}$ in the FL setting, the valuation function we use is the dual of $v_{FL}^{(t)}$, the validation accuracy of the weighted averaged model at round $t$, analogous to FedAvg [38]:
> $$v_{FL}^{(t)}(C) = acc \left( \sum_{i \in C} \theta_i^{(t)} \cdot \frac{n_i}{\sum_{j \in C} n_j} \right).$$
>
> **Experiment Setup**
>
> We conduct preliminary experiments on CIFAR-10 using FedAvg with ResNet-18 for 8 communication rounds. The training data is assigned to 3 clients as follows:
> - Client 1 receives a random half of the dataset;
> - Client 2 receives all data from 7 randomly chosen labels in the remaining half;
> - Client 3 receives the rest.
>
> Clients 1 and 2 always join at round $t=0$. We vary the joining time of client 3 and assume, following [60], that clients remain in the FL process once joined. Client 3 will train on its own data until it participates in the FL process. We set $\beta=0.98$ for a moderate penalization on late participation.
>
> **Experiment Results**
>
> We first compare the reward values of each client at each communication round when all clients join at $t=0$:
>
> |$t$|1|2|3|4|5|6|7|8|
> |:-:|:-:|:-:|:-:|:-:|:-:|:-:|:-:|:-:|
> |$r_1^{(t)}$|0.044590|0.090728|0.124992|0.155551|0.177175|0.196051|0.207676|0.216027|
> |$r_2^{(t)}$|0.025888|0.071292|0.104497|0.127404|0.143289|0.155083|0.164143|0.171478|
> |$r_3^{(t)}$|-0.020531|-0.004606|0.003255|0.014589|0.024737|0.031186|0.037289|0.041416|
>
> We observe that $r_1^{(t)}$ > $r_2^{(t)}$ > $r_3^{(t)}$, which is consistent with how we assign each client with their data: Client 1 has the most valuable data and client 3 the least. This reflects **fairness and equal-time desirability (F4)**. As we are working with non-IID setting, a client's gradient update may degrade the global model performance in early communication rounds. Hence, we can see some negative entries early on which may violate non-negativity (F1) and individual rationality (F2), but this is expected as the adapation no longer enjoys the full theoretical guarantees, and the results improve in later communication rounds/near convergence.
>
> We then investigate how delaying client 3' participation affects its reward values:
>
> |$t$|1|2|3|4|5|6|7|8|
> |:-:|:-:|:-:|:-:|:-:|:-:|:-:|:-:|:-:|
> |$r_3^{(t)}$ (joins at $t=0$)|-0.020531|-0.004606|0.003255|0.014589|0.024737|0.031186|0.037289|0.041416|
> |$r_3^{(t)}$ (joins at $t=1$)|-0.025382|-0.008411|0.001155|0.013080|0.021434|0.028065|0.034989|0.039743|
> |$r_3^{(t)}$ (joins at $t=2$)|0.000000|0.007859|0.006993|0.014581|0.022988|0.029317|0.035224|0.040227|
> |$r_3^{(t)}$ (joins at $t=3$)|0.001813|-0.096688|-0.066300|-0.041649|-0.023047|-0.007794|0.001940|0.010262|
>
> We observe that **as client 3 joins later, its reward value generally decreases**, reflecting **time-based strict monotonicity (F8)** and demonstrates that our method disincentize clients from delaying their participation. Some deviations (e.g., when client 3 joins at $t=2$) are expected, as the adaptation no longer enjoys the full theoretical guarantees. However, the overall trend remains consistent.
>
> While our adapation method demonstrates promising empirical results, we view this as a first step toward broader applicability. Future directions include (i) developing a theoretically sound method for the online FL setting, and (ii) designing mechanisms to retroactively adjust rewards to address the aforementioned "claw back rewards" challenge.
>
> Thank you for your comments! We hope our clarifications above can improve your opinion of our work.

---

> > ### Comment · Reviewer_E1vJ · 2025-08-04
> >
> > Thank you to the authors for the detailed response and additional experiments! They have answered my questions. I do not have other questions.

---

> > > ### Author Response · Authors · 2025-08-05
> > >
> > > Dear Reviewer, we are glad that our response and additional experiments addressed your questions. If you feel that the clarifications have strengthened our submission, we would greatly appreciate if you update the review accordingly. Thank you!

---

### Official Review · Reviewer_5syp · 2025-07-02

**Clarity:** 3
**Significance:** 2
**Originality:** 3
**Rating:** 4
**Confidence:** 3

**Summary:**

This paper studied the problem of data sharing in collaborative machine learning. Traditional frameworks typically assume all the parties are joining the data sharing mechanism at the same time. However, different parties may have different preparation time, and earlier joiners suffer from higher risk. Motivated by these practical concerns, this paper proposed a new framework of data sharing, taking into the time-aware incentives into consideration.

The paper proposed new methods for deciding reward values to satisfy these time-aware incentives. The, the paper demonstrated how to generate model rewards that realize the reward values and empirically illustrate the properties of the proposed methods on synthetic and real-world datasets.

**Questions:**

- Is there any redundancy among the proposed eight time-aware reward incentives? Does complying with a subset of them imply certain compliance on others?

**Ethical Concerns:**

["NO or VERY MINOR ethics concerns only"]

**Final Justification:**

The author response was helpful in addressing my concerns, I remain my score to this paper as a Weak acceptance.

**Limitations:**

yes

**Quality:**

3

**Strengths And Weaknesses:**

Strength:

- The time-aware incentives are motivated by practical concerns, and are interesting and novel considerations. It is reasonable that the data valuation and reward realization should change if parties join the data sharing process at different times due to various reasons. Thus the paper  provides explorations on this less-studied aspect.
 - The paper proposed a detailed framework with eight incentive conditions. These conditions are illustrated in detail to incorporate the time-aware considerations.
- Theoretically, the two proposed reward scheme satisfied the proposed list of incentive conditions. Empirical results further demonstrated that the two schemes comply with the incentive conditions.

Weakness:
- there is no comparison between the proposed time-aware reward scheme and classical reward schemes that do not take the time-awareness into account. Although it's shown that the proposed two schemes comply with time-aware incentives, the benefit comparing to existing non time-aware reward schemes is unclear

---

> ### Author Rebuttal · Authors · 2025-07-31
>
> We sincerely thank the reviewer for taking the time to review our work and for recognizing that our problem is motivated by practical concerns, and the claims are backed by both theoretical and empirical results. We address your questions below:
>
> > there is no comparison between the proposed time-aware reward scheme and classical reward schemes that do not take the time-awareness into account
>
> We would like to clarify here and in our revised paper that **comparisons with classical non-time-aware reward schemes are already included in our experiments** (lines 318--322). Specifically, existing methods that do not account for joining time [41, 47] can be seen as special cases of our methods when $\beta \to \infty$ and $\gamma=0$. The baselines correspond to the horizontal lines in our experiment results (e.g., Fig. 3b and Fig. 7a-b), which suggest that they assign constant rewards regardless of joining time, thereby fail to satisfy incentive F8 (time-based strict monotonicity).
>
> > Is there any redundancy among the proposed eight time-aware reward incentives? Does complying with a subset of them imply certain compliance on others?
>
> Not with additional assumptions. If we assume non-negativity of the valuation function (line 205), then F2 (individual rationality) implies F1 (non-negativity of rewards). We still include F1 as one of the incentives as it is only implied under this extra assumption in a later section (Sec. 5), and it is convention in the literature as seen in [47, 52].
>
> Additionally, in Remark 4.1, we show that F3 (equal-time symmetry) and F7 (time-based monotonicity) together imply time-based equal-value desirability, a generalization of the classical desirability incentive [47] to the time-aware setting. We did not include this as a separate incentive to avoid redundancy.
>
> Thank you for your comments! We hope our clarifications above can improve your opinion of our work.

---

### Official Review · Reviewer_acWA · 2025-07-02

**Clarity:** 4
**Significance:** 3
**Originality:** 3
**Rating:** 4
**Confidence:** 3

**Summary:**

The paper looks at time aware incentives in collaborative learning. For instance, suppose a group of agents, each having their own data-set want to jointly train a model by pooling their datasets then we call this collaborative learning. Time-awareness looks at the aspect where the agents choose to participate (or not) at different time stamps. In such a setting, the authors look at rewarding agents based on two parameters: (1) The quality of the data contributed by the agent and (2) The time at which the agents choose to participate -- Early contributions should be incentivized (while ensuring that late, high-quality data contributions are also incentivized).

To this effect, the authors set up eight desirable properties that an incentive should satisfy. Out of these, five have been established in previous works on non-time-aware fairness in collaborative learning. They introduce three new that essentially state that agents who have same quality datasets, with one agent joining before the other agent, the one that joins earlier should get higher reward -- provided the agent's data is not useless. They also have a property that says that if an agent's data is highly valuable (which they call necessary) this agent should get a high reward irrespective of when she joins the collaboration.

**Questions:**

1) The way Property F6 is defined, it seems that if agents i, j have the same data set that is necessary to the other agents (but not necessary to each other i.e., it suffices for either i or j to join the collaboration to get a non-zero value model) and i and j join at different times, they get an equal reward -- shouldn't a more reasonable requirement be that the i gets a higher reward?

2) On line 189, you have a sentence in bold that says that "a party get higher reward by waiting longer to curate a better data set due to F(6)..." however, F(6) is very strong and this will be true only if the data set contributed by the agent who waits longer is the only data set that provides any non-zero value. This seems rather stronger than what the statement is claiming -- simply better datasets that join later may not be compensated highly -- this will depend (in a complicated fashion) on how much "better" the data set is and how the parameters \beta (or \gamma) are tuned. Is this correct?

3) Can you elaborate more on the aspect of rewards -- as far as I understand, you are rewarding agents with models whose accuracy is proportional the the rewards you get. If this is the case then the fact that you can compute a model with any desired accuracy is a rather strong assumption and should be mentioned in the paper.

**Ethical Concerns:**

["NO or VERY MINOR ethics concerns only"]

**Final Justification:**

I have read the authors' response, they have addressed my questions. I maintain my opinion of the paper -- the technical aspects are direct but the question addressed is interesting. In the light of this, I will maintain my original score.

**Limitations:**

Yes.

**Paper Formatting Concerns:**

No concerns.

**Quality:**

2

**Strengths And Weaknesses:**

Strengths:
1) The authors have defined and addressed an important problem that has surprisingly not been studied before.
2) The set of properties the authors define are very reasonable (although I have some questions about this stated below).
3) The paper is well-written and easy to follow. I checked proofs for one section and they seemed correct to me.

Weaknesses:
From the technical side, the paper seems pretty direct. At the end of the section where the properties are defined, the authors demonstrate how Shapley value fails to provide rewards that satisfy the properties outlined (which is itself not surprising) however, their incentives are eventually based on Shapley value. For instance, one reward model is computing the Shapley value at each time point based on only the subset of agents that have joined until then and then takes a convex combination of these values. This seems a relatively straightforward extension.

However, given that the model is new, highly relevant to the real-world setting, I would say that overall I think this is a decent submission to NeurIPS.

---

> ### Author Rebuttal · Authors · 2025-07-31
>
> We sincerely thank the reviewer for taking the time to review our work and for recognizing that our problem is new and highly relevant to the real-world. We address your questions below:
>
> > Q1. ...it suffices for either i or j to join the collaboration to get a non-zero value model...
>
> We believe that there may be a misinterpretation of F6 (necessity). The precondition in F6 requires **both** i and j to be in coalition C for its value $v_C$ to be non-zero. This means that i and j are necessary to one another, and neither can achieve a non-zero value collaboration without the other. Therefore, it is appropriate to assign equal rewards to party i and j, as it ensures that both parties are incentivized to participate, regardless of who joins earlier.
>
> > Q2. ...this will depend (in a complicated fashion) on how much "better" the data set is and how the parameters \beta (or \gamma) are tuned. Is this correct?
>
> Yes, this is correct, and we appreciate the reviewer's clarification. We will revise the sentence by adding conditions under which it is true after the original bolded sentence, as follows:
>
> **A party may get a higher reward from taking time to curate a higher quality dataset instead of sharing a less valuable dataset as early as possible.** This holds in two key scenarios: (i) When a greater emphasis is placed on data quality relative to joining time, e.g., by setting a large $\beta$ or small $\gamma$. (ii) When the parties are *necessary* parties, i.e., their data is essential for achieving non-zero value collaboration.
>
> > Q3. Can you elaborate more on the aspect of rewards...
>
> We first decide a target reward value and seek to realize a model with the target reward value (Sec. 6.3). Note that **the reward value is not always measured using the accuracy function**.
>
> For example, in Fig. 3,6 and Fig. 5, we seek to control the conditional information gain. This is possible to achieve **exactly** and **efficiently** based on the likelihood tempering approach proposed in [49], which we outline in App. G.
>
> In Fig. 7, we seek to control the model validation accuracy. This is slightly harder to control and we propose to use the subset selection method (details in App. G) to achieve this **approximately**. [39] has also proposed early stopping to achieve different accuracies when training a neural network.
>
> We will add the clarification in our revision.
>
> > From the technical side, the paper seems pretty direct.
>
> We would like to clarify that while the use of Shapley value may appear straightforward in isolation, mathematically formulating all the proposed incentives in our time-aware setting (Sec. 4) is non-trivial. This includes both generalizing existing fairness notions to our time-aware setting and formalizing a new time-aware fairness perspective. Furthermore, we propose two schemes for deciding reward values (Sec. 6) that provably satisfy all the proposed incentives, this also poses additional technical challenges. We also wish to reiterate that a key contribution of our work lies in posing the problem itself. The time-aware data sharing and collaborative learning problem is underexplored despite its strong relevance to many real-world applications.
>
> > However, given that the model is new, highly relevant to the real-world setting, I would say that overall I think this is a decent submission to NeurIPS.
>
> We are encouraged that the reviewer also recognizes the importance of this problem, and we sincerely appreciate the acknowledgment.
>
> Thank you for your comments! We hope our clarifications above can improve your opinion of our work.

---

### Author Response · Authors · 2025-08-08
**Rebuttal Summary**

We thank all reviewers for their valuable time and insightful feedback. Below, we summarize the key discussions points during rebuttal:

The reviewers generally find our time-aware data sharing problem novel, interesting, and highly relevant to the real-world setting (acWA, 5syp, E1vJ, XdcU), yet underexplored (acWA, E1vJ). The proposed incentives are considered reasonable (acWA), and motivated by practical concerns (5syp). The reviewers recognize that our methods are theoretically sound with formal guarantess (E1vJ, 5syp), and the analysis is complete (XdcU), further supported by empirical validation (5syp). The reviewers also find our paper well-written and easy to follow (acWA), with clear motivation, and illustrative examples (E1vJ).

**Reviewer acWA** raised a question about the *dependence of rewards on the quality of data and hyperparameters*. We addressed their concern by clarifying in the rebuttal and including detailed explanations in our revised paper.

**Reviewer 5syp** raised concerns about *no comparison with non-time-aware reward schemes*. We clarified that such schemes are subsumed by our methods and are included as baselines in our experiments.

**Reviewer E1vJ** asked for *extension of our method to online FL setting*. We adapted our method to this setting and included additonal experiments demonstrating promising results, which showcases the broader applicability of our method.

**Reviewer XdcU** was mainly concerned about *impacts of different data distributions on our valuation methods*. We clarified that our data valution depends on the model, and our experiments include both IID and non-IID data distributions. The reviewer find this reasonable and has no other concerns.

---

### Decision · Program_Chairs · 2025-09-17

**Decision:**

Accept (poster)

**Comment:**

This paper formalizes a new problem in collaborative data sharing, where agents are rewarded differently based on the time of their participation in addition to their quality of participation. Eight incentives are proposed for this problem, with five borrowed from prior studies and three newly designed to encourage early participation. New methods are proposed to decide reward values that satisfy these incentives with theoretical guarantees. These methods are also empirically validated on synthetic and real-world data sets.

---

All reviewers find the new problem interesting and practical. Some reviewers compliment on the theoretical analysis and clear writing. All reviewers are on the positive side.

A notable concern shared by two reviewers (acWA, XdcU) is that the proposed methods are relatively straightforward. Authors defend that formalizing the problem is the key contribution and the presented incentives are non-trivial. I agree the merits outweigh the weakness on this aspect, and encourage authors to deepen this linear of research in future.

Another notable concern shared by two reviewers (5syp, XdcU) is the lack of baselines in the experiment. Authors explain the baselines of non-time aware reward schemes are compared with. Their choice look reasonable, considering the problem is newly formalized.

---

Overall, I think this is a neat and decent work to be presented at NeurIPS. Authors are encouraged to incorporate reviewers’ feedback in revision, and consider advancing the work with more depth in future.